# Learning without Isolation: Pathway Protection for Continual Learning

Zhikang Chen [* 1 2]  Abudukelimu Wuerkaixi [* 1 2]  Sen Cui [* 1]  Haoxuan Li [3]  Ding Li [1]  Jingfeng Zhang [4 2]
Bo Han [5 2]  Gang Niu [2]  Houfang Liu [1]  Yi Yang [† 1]  Sifan Yang [1]  Changshui Zhang [† 1]  Tianling Ren [† 1]

## Abstract

Deep networks are prone to *catastrophic forgetting* during *sequential task learning*, i.e., losing the knowledge about old tasks upon learning new tasks. To this end, *continual learning* (CL) has emerged, whose existing methods focus mostly on regulating or protecting the parameters associated with the previous tasks. However, parameter protection is often impractical, since the size of parameters for storing the old-task knowledge increases linearly with the number of tasks, otherwise it is hard to preserve the parameters related to the old-task knowledge. In this work, we bring a dual opinion from neuroscience and physics to CL: in the whole networks, *the pathways matter more than the parameters* when concerning the knowledge acquired from the old tasks. Following this opinion, we propose a novel CL framework, *learning without isolation* (LwI), where *model fusion* is formulated as *graph matching* and the pathways occupied by the old tasks are protected without being isolated. Thanks to the sparsity of activation channels in a deep network, LwI can adaptively allocate available pathways for a new task, realizing pathway protection and addressing catastrophic forgetting in a *parameter-efficient* manner. Experiments on popular benchmark datasets demonstrate the superiority of the proposed LwI.

## 1. Introduction

Continual learning, a scenario that requires a model to handle a continuous stream of tasks while preserving perfor-

*Equal contribution †Corresponding authors [1]Tsinghua University, Beijing, P.R.China [2]RIKEN [3]Peking University [4]The University of Auckland [5]Hong Kong Baptist University. Correspondence to: Yi Yang <yiyang@tsinghua.edu.cn>, Changshui Zhang <zcs@mail.tsinghua.edu.cn>, Tianling Ren <RenTL@tsinghua.edu.cn>.

*Proceedings of the 42nd International Conference on Machine Learning*, Vancouver, Canada. PMLR 267, 2025. Copyright 2025 by the author(s).

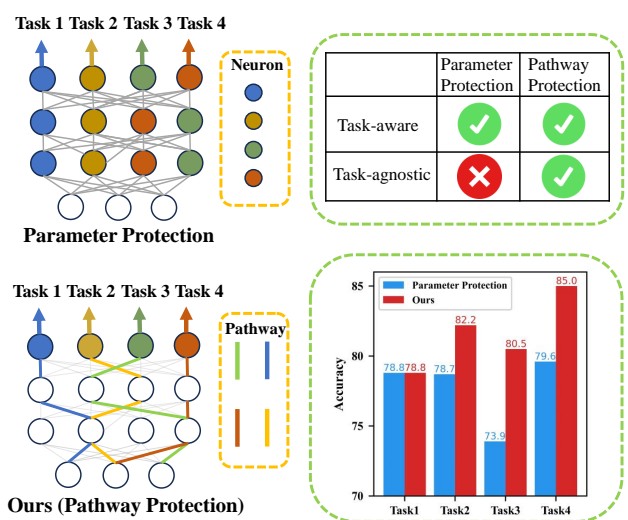

*Figure 1.* **Left Figures:** The illustrative comparison diagram between our method and the parameter-protective approach depicts the key distinctions in our methodologies. **Bottom Right Figure:** The performance comparison between our method and the WSN (Kang et al., 2022a) method. **Top Right Figure:** We showcase the ability of our method to adapt even in task-agnostic scenarios, whereas the parameter-protective approach requires knowledge of task identifiers for effective recognition.

mance on all seen tasks, is pivotal for the advancement of artificial general intelligence (Masana et al., 2022; Liang & Li, 2024; Wang et al., 2024). The approach, mirroring the human learning process of acquiring and retaining diverse experiences about the real world, confronts a significant challenge: *catastrophic forgetting* (McCloskey & Cohen, 1989). This phenomenon results in the diminished proficiency of model in prior tasks after learning on new ones.

Various continual learning approaches have been proposed to mitigate the issue of catastrophic forgetting, broadly categorized into three types. **Regularization-based approaches** entail adding regularization terms that leverage the weight information of previous tasks during the training of the current task. While this approach can mitigate catastrophic forgetting to some extent by constraining parameter shifts and ensuring protection of model parameters, it tends to result in relatively lower performance when confronted with significant variations in data characteristics. **Rehearsal-based approaches** preserve data segments

from previous tasks or use synthesized pseudo-data to retain previous knowledge while learning new tasks, which can achieve a more unified output range for the classification heads, leading to superior performance in scenarios of task agnostic. However, from the perspective of data privacy protection (Agarwal et al., 2018), this approach does not suffice. **Architecture-based approaches** focus on protecting parameters through techniques, achieving performance that matches or exceeds that of previous network training. However, they have two drawbacks, one is the requirement to know the task to which the identified object belongs in order to achieve accurate recognition, and another is that this method leads to the isolation of tasks, hindering effective communication and information sharing among them.

We argue that existing architecture-based continual learning methods do not adequately leverage the overall consideration of the sparsity of activation channels in deep networks. As illustrated in Figure 1, We adopt a holistic perspective on the deep network, allocating distinct activation pathways for each task through pathway protection involves assigning unique pathways for information transmission in the deep network. Here, **pathway** (Kipf & Welling, 2016; Zoph & Le, 2016; Huang et al., 2017; Vaswani et al., 2017) refers to the trajectories the data take through the deep network, traversing from the input layer through intermediate layers to the output layer. In this context, the concept of **channel** is akin to a neuron. The parameter-protective approach primarily involves pruning or masking operations on neurons maintains performance when task is known, it lacks consideration for the overall deep network structure. Consequently, in subsequent tasks, the reducible number of learnable parameters hinders the achievement of optimal performance. As depicted in the Figure 1, our approach is expected to outperform the latest parameter-protective methods WSN (Kang et al., 2022a). Meanwhile, considering brain's hierarchical, sparsity, and recurrent structure (Friston, 2008), brain activity relies on sparsity connections, where only a few neurons respond to any given stimulus (Babadi & Sompolinsky, 2014), brain learns and retains knowledge by re-configuring existing neurons to create more efficient neural pathways. Therefore, pathways protection is all you need.

Inspired by compensatory mechanisms observed in neuroscience and based on the sparsity of activation channels in neural networks, we propose a novel method to maintain the overall stability of deep network channels while allocating distinct pathways to different tasks across the network. The proposed approach initiates with training a model on the first task, followed by training a new model for each subsequent task. Then, a matching procedure is employed to fuse the new and old models, yielding a merged model. Conventional model fusion methods involve straightforward weight averaging (McMahan et al., 2017; Jiang et al., 2017), yet deep network parameterizations are often highly redundant, lacking one-to-one correspondence between channels (Singh & Jaggi, 2020). Simple averaging may lead to interference and even cancellation of effective components, a concern exacerbated during continual learning. Hence, in this paper, we align channels before model fusion. Several studies (Zhou et al., 2022; Hu et al., 2022; Yang et al., 2023; Gao et al., 2024) have highlighted the following characteristics of neural networks, in the shallow layers of the deep network, where tasks share more common features, we match the channels with high similarity to enhance mutual commonality. In contrast, in deeper layers, where tasks exhibit more specific characteristics, we match channels with low similarity to facilitate the fusion of distinct task features while preserving their distinctiveness, thus achieving pathway protection.

Figure 2 intuitively demonstrates the effectiveness of our approach. The concept of **"Activation level"** refers to the average magnitude of the weights obtained after activation in the last layer of the feature extraction phase. We use activation levels to measure whether pathways associated with different tasks can be distinguished. We present the activation output of data from different tasks in the last convolution layer of the trained model. As depicted in the left subplot of Figure 2, our method consistently exhibits a distinctive prominence for each task. In other words, our method adaptively allocates a set of pathways for each task, preventing the knowledge of old tasks stored in deep network parameters from being overwritten when learning new tasks, which helps mitigate catastrophic forgetting. In contrast, the Learning without Forgetting (LwF) method (Li & Hoiem, 2017) probably demonstrates nearly uniform channel activation levels for each task, leading to mixed channel utilization among tasks. As the accuracy plot in the top right corner illustrates, even after training on new tasks, our method maintains consistent or better performance on previous tasks.

It is worthwhile to summarize our key contributions as follows:

1. We explored a new direction, employing pathway protection approach for continual learning.

2. We proposed a novel data-free continual learning approach, *learning without isolation* (`LwI`), based on graph matching.

3. Our experiments on both CIFAR-100 and Tiny-Imagenet datasets demonstrate that our framework outperforms other methods. The source code of our framework is accessible at `https://github.com/chenzk202212/LwI`.

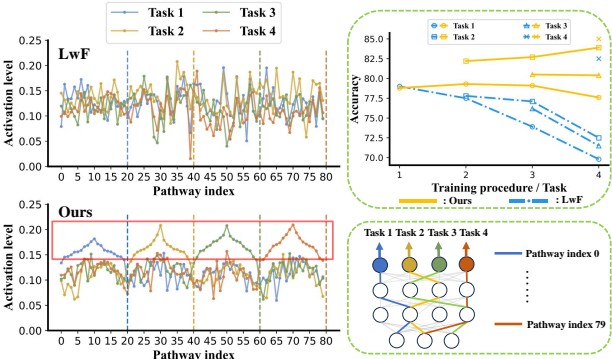

*Figure 2.* **Left Figure:** A comparison between our approach and LwF (Li & Hoiem, 2017). The activation values in the last convolution layer of the models are displayed across channels. The channels of the models have been rearranged along the horizontal axis for clearer demonstration. **Bottom Right Figure:** An explanatory legend for the horizontal axis (channel index) in the left figure. **Top Right Figure:** A comparative analysis under the condition of task awareness between our method and LwF indicates that our accuracy remains largely unchanged, contrasting with a substantial decline observed in the case of LwF.

## 2. Related Work

**Continual learning.** Deep networks exhibit a static structure, implying that once a task is learned, network parameters need to remain fixed to prevent catastrophic forgetting (Wang et al., 2024; Zhao et al., 2024). However, continual learning addresses a more prevalent scenario in which tasks arrive as a continuous data stream for the network to learn. In this context, strategies like regularization-based, rehearsal-based, and dynamic architecture-based approaches are employed to mitigate catastrophic forgetting. Regularization-based methods apply constraints to limit changes in weights or nodes from past tasks, thereby reducing catastrophic forgetting. For instance, methods like EWC (Kirkpatrick et al., 2017) incorporate the Fisher information matrix of previous task weights, while RWalk (Chaudhry et al., 2018) merges this matrix's approximation with online path integration to gauge parameter importance. LwF method, on the other hand, employs output alignment to prevent the model weights from a large shift. SPG (Konishi et al., 2023) employs the Fisher information matrix to control the updates of each parameter, enabling more granular parameter protection. Rehearsal-based approaches involve preserving portions of data from previous tasks or using some techniques to generate pseudo-data (Shin et al., 2017). This data is then combined with the current dataset during the training for the next task, alleviating catastrophic forgetting. For example, both approaches, LUCIR (Hou et al., 2019) and iCaRL (Rebuffi et al., 2017), leverage the technique of preserving a portion of previously acquired data along with knowledge distillation for incremental learning. Continual Prototype Evolu-

tion (CoPE) (De Lange et al., 2021) combines the principles of the nearest-mean classifier with a reservoir-based sampling strategy. Dynamic architecture-based methods encompass expanding models and employing parameter isolation techniques to retain previous knowledge while accommodating new knowledge expansion.

**Parameter isolation-based continual learning.** This approach aims to safeguard parameters to preserve knowledge acquired from previous tasks (Zhang et al., 2024b). The Piggyback method (Mallya et al., 2018) involves learning a series of masks over a post-pretrained model, corresponding to various tasks, resulting in a series of task-specific subnetworks. The PackNet method (Mallya & Lazebnik, 2018) uses pruning method to protect neurons, which are important to previous tasks. CLNP method (Golkar et al., 2019) divides neurons in the deep network into active, inactive, and interference parts, utilizing previously learned features and unused weights from the network to train new tasks. Supsup method (Wortsman et al., 2020) employs masking to protect specific parameters important for tasks. Chen et al. (2020) prunes the model to obtain the optimal subnetwork for the task, thus preserving knowledge and achieves generalization for new tasks through re-growing. GPM (Saha et al., 2021) utilizes gradient mapping to project the knowledge from previous tasks into mutually orthogonal gradient subspaces, thereby enabling continual learning. The WSN algorithm (Kang et al., 2022a), based on the lottery hypothesis, learns a compact subnetwork for each task while maintaining the weights chosen for previous tasks unchanged. SPU (Zhang et al., 2024a) employs causal tracking to select model parameters for updates, thereby facilitating knowledge protection. However, most of these methods involve pruning or masking based on network weights, leading to non-structured modifications that risk compromising the integrity of network. Our approach integrates the channel properties of network, which allows different tasks to utilize distinct pathways for propagation and flow, preserving the overall integrity of the deep network without causing disruption.

**The sparsity of deep network.** According to the mechanisms observed in neuroscience, in the brains of healthy adults, the density of connections remains roughly constant. Despite learning more tasks, the capacity of neurons in the brain remains relatively unchanged. Meanwhile, within deep networks, this phenomenon also manifests. Upon completion of training, deep networks typically exhibit sparse activation, with a small proportion of effectively activated neurons (Han et al., 2015; Liu et al., 2015; Fan et al., 2020; Dai et al., 2021). According to the findings of Mao et al. (2017), deep networks exhibit an inverse relationship between overall accuracy and granularity. Specifically, under similar sparsity levels, increased granularity is associated with improved accuracy. Under comparable sparsity con-

ditions, finer granularity tends to yield optimal accuracy. Concurrently, the MEMO method (Zhou et al., 2022) highlights similarities in the shallow layers of different models while showcasing differences in the deeper layers. Therefore, we hypothesized that within the coarser granularity (shallow layers) of the deep network, a denser occupation of channels occurs, while in the finer granularity (deeper layers), channel occupation tends to be sparser.

## 3. Preliminary

In this section, we provide an elucidation of the problems to be addressed and the prerequisite knowledge required for subsequent methods. In Section 3.2, we present an exposition on continual learning. Sections 3.1 and 3.3 introduce the foundational knowledge underpinning our approach, which includes deep network sparsity and graph matching algorithms.

Consider a supervised continual learning scenario where learners need to solve $T + 1$ tasks in sequence without catastrophically forgetting old tasks. At the same time, due to data privacy restrictions, we cannot store data from previous tasks. We use $D_{t+1} = \{X_{t+1}; Y_{t+1}\}$ to denote a dataset for task $t + 1$. $X_{t+1} = \{x_1, ..., x_n\}$ and $Y_{t+1} = \{y_1, ..., y_n\}$ represent that the dataset includes n data classes along with their corresponding labels for task $t + 1$. And we use $M_t$ to denote a trained model for task $t$. Meanwhile, $D_{1:t} = \{X_1, ..., X_t; Y_1, ..., Y_t\}$ denotes datasets for all seen tasks from task 1 to task $t$. We represent the deep network model using the following formula:

$$M_t(x) = f(\theta), \tag{1}$$

and a standard continual learning scenario designed to learn a series of tasks by minimizing optimization problems at each step:

$$\min_{\theta} L(f(X_{t+1}; \theta), Y_{t+1}), \tag{2}$$

where $L$ denotes the loss function used when training task $t + 1$. It is well known that simply optimizing the loss function can easily lead to catastrophic forgetting.

### 3.1. Graph matching for deep network fusion.

Recently, some studies have employed graph matching approaches for model fusion (Su et al., 2021). Graph matching bears resemblance to a *quadratic assignment problem* (QAP) (Loiola et al., 2007), with the objective of establishing correspondences between the nodes in an image and the edges connecting these nodes. The activation distribution of deep network channels is not fixed across training iterations, resulting in some neurons exhibiting high activation for one task, but low activation for another. If a straightforward averaging fusion is performed, it may lead to interference and blending of effective components within

the deep network (Singh & Jaggi, 2020). Hence, aligning the channels before fusion becomes a crucial step in the integration process.

In this context, we conceptualize the matching process between deep networks as a graph matching problem. In our framework, a deep network is conceptualized as an image. This representation enables the alignment of two deep networks through the application of a graph matching algorithm. At each layer, we interpret the channels within that layer as nodes in an image, and the connections between adjacent layer channels as edges. It is noteworthy that, within deep networks, we assert that matching occurs exclusively within each layer, as cross-layer matching holds no significant relevance. This approach facilitates the effective application of graph matching methods in deep networks, given their large-scale neuron configuration. The specific formula for graph matching is presented as follows:

$$\max_{\boldsymbol{P}} \sum_{a=0}^{N-1} \sum_{b=0}^{N-1} \sum_{c=0}^{N-1} \sum_{d=0}^{N-1} \boldsymbol{P}_{a,b} \boldsymbol{K}_{[a,c,b,d]} \boldsymbol{P}_{c,d},$$
$$\text{s.t.} \quad \boldsymbol{P}_0 = \boldsymbol{I}; \quad \boldsymbol{P}_L = \boldsymbol{I};$$
$$\sum_{a=0}^{N-1} \boldsymbol{P}_{[a,c]} = 1, \quad \forall m \in [1, L-1]; \tag{3}$$
$$\sum_{c=0}^{N-1} \boldsymbol{P}_{m[a,c]} = 1, \quad \forall m \in [1, L-1].$$

where $a$ and $c$ represent node indices between adjacent layers in $modelX$ as shown in Figure 3.2, $b$ and $d$ represent node indices between adjacent layers in $modelY$ as shown in Figure 3.2, $L$ represents the index of the last layer in the neural network, $N = \sum_{m=0}^{L} N_m$ represents the sum of the number of nodes across all layers, $N_m$ represents the number of nodes across layer $m$, $\boldsymbol{K}$ represents the similarity matrix between adjacent layers in $modelX$ and $modelY$, $\boldsymbol{P}_0$ represents the permutation matrix for the first layer, $\boldsymbol{P}_N$ represents the permutation matrix for the last layer, $\boldsymbol{P}_m, m \in [1, N-1]$ denotes the permutation matrix for intermediate layers. We need to solve the assignment matrix $\boldsymbol{P}$, and according to the formula, we can find that the time complexity of using the graph matching method is $O(N^4)$. However, based on the above analysis, we use a layer-by-layer calculation of the assignment matrix in this paper to align the channels at each layer of the deep network, so that $N$ is not the number of all channels, but the number of channels in each layer. More analysis could be found in the appendix A.3.

### 3.2. Problem Statement

In practical applications, highly precise matching results are not necessary, and the majority of current work focuses on the approximate matching of nodes or edges. The previous

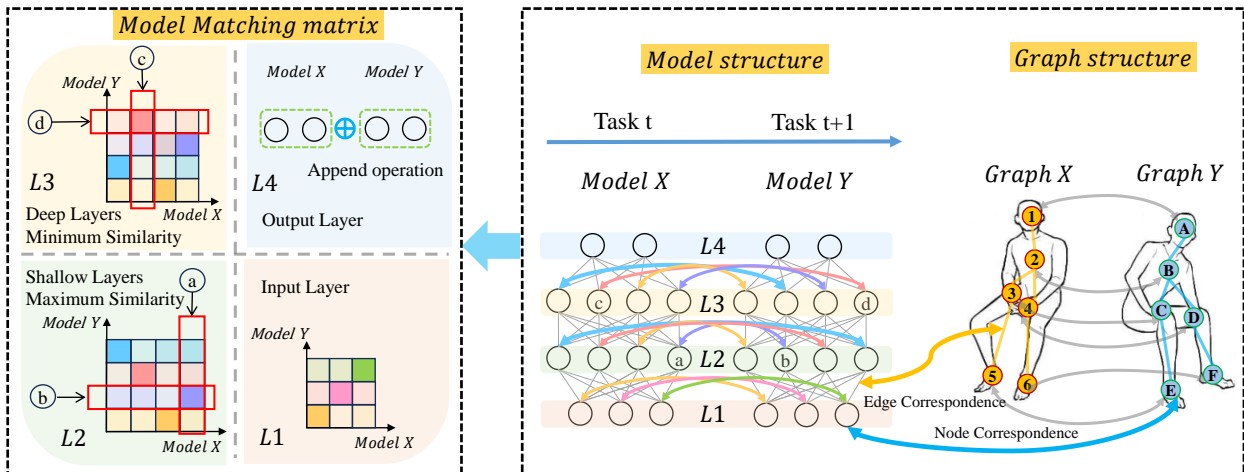

*Figure 3.* The overall structure of our proposed LwI algorithm. In the **right diagram**, we represent the deep network in four parts: L1 corresponds to the input layer, L2 to the shallow layers, L3 to the deeper layers, and L4 to the output layer. The channels in the deep network can be analogous to nodes in a graph, and the connections between channels correspond to the edges in the graph. On the **left side**, L1 requires no matching operation. L4 only needs to append operations for the output heads of different tasks. L2 matches the channels with maximum similarity. Conversely, L3 undergoes minimization of similarity matching.

work can be divided into classical methods and deep graph matching methods. In this paper, a more commonly used method, Sinkhorn algorithm (Cuturi, 2013), is used. The Sinkhorn algorithm, rooted in entropy regularization, transforms a binary 0-1 matrix into a soft matching matrix with a sum of 1 through a process of bi-directional relaxation.

### 3.3. The sparsity of deep network.

The primary rationale behind this approach stems from the sparsity of deep networks. To accommodate future learning tasks, continual learners often utilize over-parameterized deep networks. The reason is that continual learning frequently relies on over-parameterized deep deep networks to allow flexibility for future tasks.

We believe that a deep network is composed of multiple layers, and we use $\ell$ to represent the index of one layer of the deep network. Meanwhile, it has been observed in previous studies (Zhou et al., 2022) that shallow layers across models of different tasks exhibit notable similarities, whereas deeper layers demonstrate distinct characteristics. The deep network model can be decoupled into a classifier, denoted as $G(\cdot)$, and a feature extractor, represented as $F(\cdot)$. The feature extractor further bifurcates into a shallow-layer deep network $S(\cdot)$ and a deep-layer deep network $D(\cdot)$ in Eq.(4):

$$M(x) = G_\ell(F_\ell(x)) = G_\ell(D_\ell(S_\ell(x))). \quad (4)$$

The Lottery Ticket Hypothesis (Frankle & Carbin, 2018) posits that within deep networks, there exist specific "winning tickets" generated during the training process. These winning tickets, it suggests, enable comparable performance

with the entire network in different tasks while employing fewer parameters and requiring shorter training times. Regarding the sparsity aspect of deep networks, the Lottery Ticket Hypothesis asserts that only a small subset of connections (weights) within the network is crucial for learning and performance, while the remaining connections can be pruned (set to zero) without significantly affecting the performance of network.

Previous studies show that deep networks are sparse—not all parameters need computation, as many are inactive. By assigning different channels to different tasks, we can leverage this sparsity to match or even surpass performance while mitigating catastrophic forgetting in continual learning.

## 4. Methodology

In this section, we delve into a comprehensive discussion of our proposed continual learning structure based on graph matching, along with its specific implementation. Section 4.1 outlines the framework of proposed method for continual learning based on graph matching. We fuse the model trained on a new task with the one trained on previous tasks. Rather than merely averaging the parameters of the two models, we conduct pathway alignment based on graph matching before fusion. To achieve the protection and sharing of knowledge, we employ different similarity matrix across different layers of the deep network. Section 4.2 provides a detailed account of the optimization process.

## 4.1. Graph-not-Matching for Continual Learning

Unlike previous architecture-based methods, which usually mask weights crucial for the previous task, our strategy stems from the holistic nature and sparsity of neural networks. We believe that safeguarding previous tasks through misaligned addition represents a matching approach. This method not only protects the overall integrity of the deep network, but also allocates different directional channels for various tasks.

**Overview.** The overall structure of our proposed LwI algorithm is shown in Figure 3. When a new task arrives, we train a new model for it. We then employ graph matching for channel alignment before model fusion, by analogizing neural network nodes to nodes in graph matching and connections between deep network channels to edges in graph matching. The specific alignment operations, as illustrated in the left diagram, involve matching the channels with high similarity in shallow layers and low similarity in deep layers. Meanwhile, matching operations are not required in L1, and in L4, only append operation is necessary. The overall process of our proposed method is illustrated in Algorithm 3 in appendix.

**Model Fusion Process**. The fusion process of the new and old models is illustrated in Algorithm 1. We use Euclidean Distance in Eq.(5) to compute distances between weights and subsequently employ specific graph matching algorithms:

$$\boldsymbol{K}_{[a,c,b,d]} = \|e_{ac} - e_{bd}\|_2, \tag{5}$$

where $\boldsymbol{K}$ denoted the similarity matrix, $e_{ac}$ and $e_{bd}$ denote the similarity relationship between the channels of adjacent layers in two models, specifically focusing on the edges $(a, c)$ and $(b, d)$, respectively. We also utilize cosine similarity for measurement, and specific details can be found in the ablation study and the appendix C.6. In the shallow layers, we need to maximize similarity matching for protecting old knowledge and promoting the collaboration of different tasks. Hence, the similarity matrix between two edges is itself. Conversely, in the deeper layers, we repeat a similar process but utilize a minimizing similarity approach for protecting the individuality of different tasks, facilitating misaligned fusion of channels. Therefore, the similarity matrix between two edges is represented by its negation. The specific matching process can be found in Algorithm 2 in the appendix. This process yields a permutation matrix, enabling us to perform matrix multiplication between the old model and the permutation matrix. Subsequently, the permutation matrix of similarity from the previous layer is multiplied with the parameter matrix of the current layer, ensuring the coherence of the connections between the channels. The matrix multiplication of the permutation matrix for the current layer is performed with itself, positioning

---

**Algorithm 1** Model Fusion Process

**Input:** the weight matrix between the layer $l-1$ and $l$ is denoted as $\boldsymbol{W}^{(l-1,l)}$, $\boldsymbol{P}^{(l-1,l)}$ represents the corresponding permutation matrix, fusion coefficient is $k$.
**Output:** the fusion model $\boldsymbol{W}_{fusion}$.
**for** $layer$ $1, ..., N$ **do**
 Calculate the permutation matrix $\boldsymbol{P}^{(l-1,l)}$ according to the Algorithm 2 in appendix;
 **if** $layer == 1$ **then**
  Calculate $\widehat{\boldsymbol{W}}_o^{(0,1)} \leftarrow \boldsymbol{P}^{(0,1)\top} \boldsymbol{W}_o^{(0,1)}$;
 **end**
 **else**
  Calculate $\widetilde{\boldsymbol{W}}_o^{(l-1,l)} \leftarrow \boldsymbol{W}_o^{(l-1,l)} \boldsymbol{P}^{(l-2,l-1)}$;
  Calculate $\widehat{\boldsymbol{W}}_o^{(l-1,l)} \leftarrow \boldsymbol{P}^{(l-1,l)\top} \widetilde{\boldsymbol{W}}_o^{(l-1,l)}$;
 **end**
 $\boldsymbol{W}_{fusion}^{(l-1,l)} = k * \widehat{\boldsymbol{W}}_o^{(l-1,l)} + (1-k) * \boldsymbol{W}_n^{(l-1,l)}$;
**end**
$\boldsymbol{W}_o = \boldsymbol{W}_{fusion}$;

---

the most similar or dissimilar channels accordingly. This process achieves channel alignment within the current layer.
**Graph-Matching and Graph-not-Matching**. We adopted the combined approach of maximizing and minimizing similarities for the following reasons: **1).** To facilitate collaboration between tasks. **2).** Considering the sparsity of deep network channels, we allocate different channels for different tasks in the sparse layer, thereby preserving the characteristics of each task. The key to implementing soft matching in our method lies in calculating the optimal transport matrix, which is the matching matrix $\boldsymbol{P}$. Here, we provide a more detailed explanation of Algorithm 1. Our goal is to use the similarity matrix $\boldsymbol{K}$ to obtain the matrix $\boldsymbol{P}$, where $\boldsymbol{P}_{ab}$ represents the optimal amount of mass to transport the $a$-th neuron in the $l$-th layer of $model X$ to the $b$-th neuron in the $l$-th layer of $model Y$. The implementation process is that, in the shallow layers, we observe that different tasks occupy denser channels with shared features. Consequently, for these distinct tasks, we consolidate their most similar channels, facilitating mutual reinforcement of common features, for the collaboration of knowledge among different tasks. Meanwhile, in the deeper layers, we observe that different tasks occupy sparser channels, emphasizing distinct characteristics. Thus, for these tasks, we consider the misaligned fusion of channels that represent unique traits of each task, aiming to safeguard the individual characteristics.

## 4.2. Optimization

Knowledge distillation aims to mitigate semantic discrepancies between the new and old models, otherwise, model fusion loses its significance. Additionally, in training a

*Table 1.* Task-agnostic and Task-aware accuracy (%) of different methods. Our approach is based on data-free, but the results of exemplar-based methods are also provided.

| Dataset | Architecture | Method | Exemplar | Task-agnostic | | | Task-aware | | |
|---|---|---|---|---|---|---|---|---|---|
| | | | | 5 splits | 10 splits | 20 splits | 5 splits | 10 splits | 20 splits |
| CIFAR-100 | ResNet32 | EWC | no | $31.81 \pm 1.45$ | $21.14 \pm 0.98$ | $12.32 \pm 0.56$ | $64.22 \pm 0.83$ | $65.86 \pm 1.55$ | $63.43 \pm 1.59$ |
| | | RWalk | | $21.40 \pm 1.22$ | $20.07 \pm 1.91$ | $12.49 \pm 1.36$ | $64.98 \pm 0.97$ | $69.16 \pm 1.29$ | $67.98 \pm 1.38$ |
| | | LwF | | $37.54 \pm 0.43$ | $25.78 \pm 0.43$ | $15.86 \pm 1.15$ | $74.63 \pm 0.72$ | $75.98 \pm 1.03$ | $76.37 \pm 1.44$ |
| | | SPG | | $30.74 \pm 0.27$ | $22.54 \pm 1.23$ | $11.28 \pm 0.22$ | $62.22 \pm 1.54$ | $70.34 \pm 0.52$ | $72.39 \pm 0.05$ |
| | | SPU | | $34.56 \pm 0.93$ | $23.44 \pm 0.36$ | $17.33 \pm 0.21$ | $66.02 \pm 0.47$ | $73.31 \pm 0.21$ | $78.34 \pm 0.47$ |
| | | GPM | | - | - | - | $71.72 \pm 0.35$ | $78.74 \pm 1.17$ | $80.47 \pm 0.33$ |
| | | WSN | | - | - | - | $75.47 \pm 0.48$ | $80.12 \pm 0.60$ | $82.51 \pm 0.50$ |
| | | **Ours** | | $\textbf{43.42} \pm \textbf{0.58}$ | $\textbf{30.62} \pm \textbf{1.08}$ | $\textbf{20.31} \pm \textbf{0.77}$ | $\textbf{76.10} \pm \textbf{0.33}$ | $\textbf{81.12} \pm \textbf{0.90}$ | $\textbf{83.19} \pm \textbf{0.35}$ |
| | | iCaRL | 2000 | $37.23 \pm 0.74$ | $36.88 \pm 2.33$ | $33.88 \pm 3.03$ | $62.98 \pm 0.79$ | $73.40 \pm 1.46$ | $81.74 \pm 1.65$ |
| | | LUCIR | | $\textbf{48.48} \pm \textbf{1.16}$ | $\textbf{41.10} \pm \textbf{1.98}$ | $\textbf{36.46} \pm \textbf{1.83}$ | $75.40 \pm 0.57$ | $80.05 \pm 1.00$ | $\textbf{84.95} \pm \textbf{0.99}$ |
| CIFAR-100 | ResNet18 | EWC | no | $30.84 \pm 0.27$ | $18.66 \pm 0.62$ | $9.21 \pm 0.25$ | $61.25 \pm 0.46$ | $56.53 \pm 1.84$ | $51.34 \pm 0.72$ |
| | | RWalk | | $38.81 \pm 2.08$ | $21.78 \pm 0.53$ | $7.82 \pm 1.07$ | $69.41 \pm 1.70$ | $61.91 \pm 0.62$ | $57.57 \pm 1.16$ |
| | | LwF | | $44.66 \pm 0.97$ | $30.41 \pm 0.82$ | $16.66 \pm 1.36$ | $79.96 \pm 0.52$ | $81.35 \pm 0.51$ | $81.45 \pm 0.67$ |
| | | SPG | | $26.32 \pm 0.57$ | $20.16 \pm 1.51$ | $10.54 \pm 0.14$ | $64.98 \pm 0.97$ | $69.16 \pm 1.29$ | $67.98 \pm 1.38$ |
| | | SPU | | $43.79 \pm 0.40$ | $25.12 \pm 0.48$ | $16.08 \pm 0.71$ | $74.63 \pm 0.72$ | $75.98 \pm 1.03$ | $76.37 \pm 1.44$ |
| | | GPM | | - | - | - | $78.23 \pm 1.13$ | $81.42 \pm 1.43$ | $86.21 \pm 0.46$ |
| | | WSN | | - | - | - | $78.65 \pm 1.33$ | $83.08 \pm 1.57$ | $86.10 \pm 0.25$ |
| | | **Ours** | | $\textbf{51.95} \pm \textbf{0.56}$ | $\textbf{36.36} \pm \textbf{1.06}$ | $\textbf{22.99} \pm \textbf{0.39}$ | $\textbf{81.10} \pm \textbf{0.80}$ | $\textbf{84.90} \pm \textbf{0.36}$ | $\textbf{86.49} \pm \textbf{0.55}$ |
| | | iCaRL | 2000 | $49.44 \pm 0.78$ | $39.27 \pm 0.37$ | $28.48 \pm 1.57$ | $73.84 \pm 0.36$ | $76.63 \pm 0.62$ | $78.49 \pm 0.74$ |
| | | LUCIR | | $\textbf{55.67} \pm \textbf{1.04}$ | $\textbf{42.56} \pm \textbf{0.97}$ | $\textbf{33.84} \pm \textbf{1.95}$ | $\textbf{81.22} \pm \textbf{0.25}$ | $\textbf{84.41} \pm \textbf{0.22}$ | $86.19 \pm 0.25$ |
| Tiny-Imagenet | ResNet18 | EWC | no | $19.21 \pm 0.31$ | $10.32 \pm 0.29$ | $4.69 \pm 0.39$ | $42.84 \pm 0.54$ | $36.21 \pm 1.07$ | $30.82 \pm 2.06$ |
| | | RWalk | | $21.69 \pm 0.64$ | $12.94 \pm 0.38$ | $7.84 \pm 0.21$ | $55.67 \pm 1.27$ | $56.14 \pm 0.29$ | $59.58 \pm 0.40$ |
| | | LwF | | $26.76 \pm 0.50$ | $20.14 \pm 0.28$ | $13.09 \pm 0.24$ | $59.66 \pm 0.47$ | $63.52 \pm 0.57$ | $70.59 \pm 0.47$ |
| | | SPG | | $22.80 \pm 0.26$ | $12.03 \pm 0.73$ | $7.86 \pm 0.24$ | $54.50 \pm 0.47$ | $57.81 \pm 0.23$ | $59.67 \pm 0.44$ |
| | | SPU | | $25.50 \pm 0.40$ | $19.98 \pm 0.06$ | $13.44 \pm 0.18$ | $57.15 \pm 0.31$ | $59.93 \pm 0.08$ | $63.64 \pm 0.38$ |
| | | GPM | | - | - | - | $58.45 \pm 0.38$ | $63.17 \pm 0.24$ | $70.16 \pm 0.42$ |
| | | WSN | | - | - | - | $57.38 \pm 0.51$ | $64.12 \pm 0.43$ | $71.54 \pm 0.43$ |
| | | **Ours** | | $\textbf{34.33} \pm \textbf{0.51}$ | $\textbf{26.15} \pm \textbf{0.22}$ | $\textbf{15.59} \pm \textbf{0.84}$ | $\textbf{62.97} \pm \textbf{0.14}$ | $\textbf{68.67} \pm \textbf{0.36}$ | $\textbf{72.74} \pm \textbf{0.27}$ |
| | | iCaRL | 2000 | $28.81 \pm 0.14$ | $23.37 \pm 0.24$ | $14.68 \pm 0.35$ | $56.17 \pm 0.34$ | $59.49 \pm 0.91$ | $61.00 \pm 0.67$ |
| | | LUCIR | | $30.17 \pm 0.37$ | $20.15 \pm 0.63$ | $13.48 \pm 0.60$ | $60.25 \pm 0.38$ | $65.52 \pm 0.16$ | $66.56 \pm 0.66$ |

model for a new task, leveraging the universally applicable knowledge from the old task model, such as shallow-level enhances the efficiency of learning through distillation. To leverage prior task knowledge, we employed previous models as pre-trained models, integrating their parameters into the current model for subsequent task training. Simultaneously, throughout the entire training process, the feature extractor of the classifier undergoes continuous modifications. If there is a noticeable drift in the feature space of the classifier, the knowledge memorized by the model may become outdated. Consequently, it is imperative to maintain a relative consistency in the feature space of the classifier during the training process. Further details can be found in the appendix C.7 and C.8.

# 5. Results and Discussion

In the main text, we present the results of three experiments, including the application of the ResNet32 architecture to the CIFAR-100 dataset, and ResNet18 to both the CIFAR-100 and Tiny-ImageNet datasets. The remaining experimental results are included in the appendix C.

## 5.1. Settings

**Datasets.** Following the work (Masana et al., 2022), we evaluate different methods on benchmark datasets with settings, including CIFAR-100 and Tiny-Imagenet datasets. Under the condition of continual learning, we use three task-splitting settings: 5 splits, 10 splits, and 20 splits.

**Architecture.** In order to verify our proposed method can achieve knowledge protection for different tasks, we conducted a large number of experiments to study the effect of model size on performance. In this article, we use ResNet32 and ResNet18 architectures (He et al., 2016) for comparison(the sizes and parameter counts of the two models are detailed in the appendix B.3).

**Baselines.** In order to demonstrate the effectiveness of our approach, we conduct comparative tests against different continual learning methods. Specifically, baseline methods include regularization-based frameworks, like EWC (Kirkpatrick et al., 2017), LwF (Li & Hoiem, 2017), RWalk (Chaudhry et al., 2018) and SPG (Konishi et al., 2023), architecture-based framework, like GPM (Saha et al., 2021), WSN (Kang et al., 2022a) and SPU (Zhang et al., 2024a), which is inapplicable in scenarios where task is unknown, and some classical rehearsal-base methods, such as LUCIR (Hou et al., 2019) and iCaRL (Rebuffi et al., 2017).

**Implementation Details.** We trained the model for 200 epoches and optimized it in conjunction with SGD, setting the batch size of the dataset as 64. For rehearsal-based methods, we set 2000 exemplars using the herding method to select (Masana et al., 2022). In addition, we evaluate the methods on task-aware and task-agnostic settings. More experimental details could be found in the appendix B.

## 5.2. Different deep network architectures on CIFAR-100 dataset.

The performance of all methods on the same dataset, that is CIFAR-100 dataset, is shown in the Table 1. Our approach surpasses the baseline performance of all without exemplar in the comparative experiments. Furthermore, when compared to methods employing exemplar such as iCaRL and LUCIR, our approach exhibits superior performance across the majority of test results.

*Table 2.* Task-agnostic accuracy (%) of methods on using minimum similarity matching on different layers.

| Method | Task-agnostic | | |
|---|---|---|---|
| | 5 splits | 10 splits | 20 splits |
| **Ours** | **43.42 ± 0.58** | **30.62 ± 1.08** | **20.31 ± 0.77** |
| Ours 2layers | 26.84 ± 0.86 | 23.50 ± 0.30 | 16.05 ± 0.28 |
| Ours 3layers | 20.22 ± 1.08 | 19.71 ± 0.82 | 13.53 ± 0.50 |
| Ours 4layers | 15.91 ± 0.95 | 13.69 ± 0.58 | 10.42 ± 0.39 |

*Table 3.* Task-aware accuracy (%) of methods on using minimum similarity matching on different layers.

| Method | Task-aware | | |
|---|---|---|---|
| | 5 splits | 10 splits | 20 splits |
| **Ours** | **76.10 ± 0.33** | **81.12 ± 0.90** | **83.19 ± 0.35** |
| Ours 2layers | 61.92 ± 0.66 | 71.30 ± 0.83 | 75.42 ± 0.44 |
| Ours 3layers | 51.01 ± 0.92 | 66.49 ± 0.37 | 71.14 ± 0.85 |
| Ours 4layers | 42.09 ± 1.45 | 55.27 ± 0.41 | 67.48 ± 0.81 |

As the network capacity increases, our performance in task-agnostic scenarios improves significantly. This is primarily attributed to the fact that, under the conditions of smaller network models, channels are more densely occupied by various tasks. As the size of the network model increases, the sparsity of the occupied channels increases.

## 5.3. Different datasets based on ResNet18 architecture.

The performance of all methods in the same deep network architecture is shown in the latter two blocks in Table 1. With the escalation of dataset complexity, channels within the same structured deep network are more extensively leveraged. Consequently, judiciously preserving channels occupied by different tasks becomes essential to achieve better performance under task-agnostic conditions.

## 5.4. Experimental testing of forgetting rates for different methods.

The experimental results for testing forgetting rates show that we used the ResNet18 architecture to evaluate forgetting rates on the CIFAR-100 dataset. The Figure 4 indicates that we achieve lower forgetting rates, and our method also demonstrates improved learning capabilities.

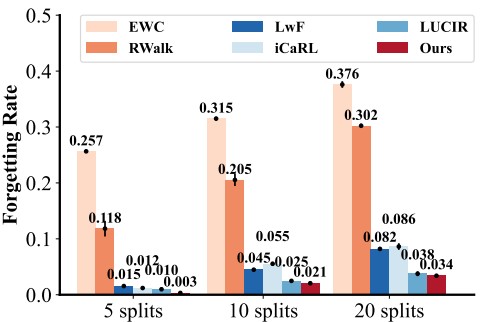

*Figure 4.* Task-aware forgetting rates of different methods.

## 5.5. Ablation Studies

To validate the effectiveness of different modules in our proposed method, LwI, we conducted ablation experiments on the model. In this context, "w/o task diversion" signifies match the channels with high similarity for every layer of the deep network, while "Ours n layers" indicates applying minimization of similarity matching for the different n layers. The term "with cosine" indicates employing cosine similarity for channel similarity measurement. More experimental details and results are available in appendix C.

**Minimum similarity matching on different layers.** The results show the effectiveness of minimizing similarity matching in the final layer.

*Table 4.* Task-agnostic accuracy (%) of methods on the validation of using task diversion module and similarity measurements.

| Method | Task-agnostic | | |
|---|---|---|---|
| | 5 splits | 10 splits | 20 splits |
| **Ours** | **34.33 ± 0.51** | **26.15 ± 0.22** | **15.59 ± 0.84** |
| Ours w/o task diversion | 30.75 ± 0.43 | 21.29 ± 0.34 | 14.30 ± 0.26 |
| Ours with cosine | 34.17 ± 0.54 | 25.98 ± 0.25 | 15.48 ± 0.65 |

**Effectiveness with task diversion module.** Using minimization of similarity matching in the final layer, facilitating channel diversion for task segregation and consequently ensuring protection across distinct tasks.

**Different similarity measurement methods.** When measuring model similarity for the purpose of model fusion, the use of Euclidean distance yields slightly higher performance compared to cosine similarity.

*Table 5.* Task-aware accuracy (%) of methods on the validation of using task diversion module and similarity measurements.

| Method | Task-aware | | |
|---|---|---|---|
| | 5 splits | 10 splits | 20 splits |
| **Ours** | **62.97 ± 0.14** | 68.67 ± 0.36 | **72.74 ± 0.27** |
| Ours w/o task diversion | 62.12 ± 0.26 | 66.94 ± 0.50 | 71.72 ± 0.60 |
| Ours with cosine | 62.69 ± 0.32 | **68.86 ± 0.24** | 72.30 ± 0.47 |

## 6. Conclusion

This paper proposes a framework for continual learning, `LwI`, achieving pathway protection between different tasks using model fusion approach. We validated our approach using two network structures of different sizes, and further validation can be performed on larger models. Our method acknowledge some limitations, notably the lack of validation of the proposed method using large models. Additionally, the graph matching algorithm can be accelerated in future work by employing sparse matrix techniques, we will investigate more effective and efficient matching processes in future work. We hope this work opens the new direction for future research, pathway protection for continual learning.

## Impact Statement

We propose a novel pathway protection-based continual learning approach. Our method is under the condition of data-free, which has significant implications for data privacy protection. The introduction of a novel method in our research represents a significant technological advancement. In future work, this innovation can potentially improve the performance of *Large Language Model* (LLM) under the circumstance of streaming tasks.

## Acknowledgement

This work was supported by the National Key Research and Development Program of China (2022YFB3204100), Scientific and Technological Innovation Project of China Academy of Chinese Medical Sciences (ZN2023A01, CI2023C002YG), China Institute for Rural Studies (CIRS2025-9), and Institute for Intelligent Healthcare, Tsinghua University. BH was supported by the NSFC General Program No. 62376235, GDST Basic Research Fund Nos. 2022A1515011652 and 2024A1515012399. Sen Cui would like to acknowledge the financial support received from Shuimu Tsinghua scholar program.

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

# A. Theoretical Supports

## A.1. Analysis

We analyze one layer of deep network channel, and first-order Taylor expansion is used for analysis (Kang et al., 2022b):

$$\mathcal{L}\left(G_\ell\left(Z'_\ell\right), y\right) \approx$$
$$\mathcal{L}\left(G_\ell\left(Z_\ell\right), y\right) + \sum_{c=1}^{C_\ell} \left\langle \nabla_{Z_{\ell,c}} \mathcal{L}\left(G_\ell\left(Z_\ell\right), y\right), Z'_{\ell,c} - Z_{\ell,c} \right\rangle_F . \quad (6)$$

Based on the above , We find that the first-order term is a deviation due to the deviation of the channel $c$, so we need to use some ways to reduce this deviation. Naturally, we think about whether we can make full use of the information of different channels brought by different tasks, so that different tasks can occupy different channels to minimize inter-task interference.

## A.2. Some methods related to graph matching

The classical methods mainly include the path-following strategy (Zaslavskiy et al., 2008), graduated assignment algorithm (Gold & Rangarajan, 1996), spectral matching algorithm (Leordeanu & Hebert, 2005), random-walk algorithm (Cho et al., 2010) and sequential Monte Carlo sampling (Leordeanu et al., 2012). The method of deep graph matching (Yu et al., 2019) has also received more and more attention in recent years.

The specific implementation of graph matching is illustrated in the following diagram5. Assuming that the nodes in graph X are labeled from 1 to 6, and the nodes in graph Y are labeled from A to F, the similarity matrix for pairwise nodes is shown in the upper right corner. Meanwhile, nodes are interconnected, forming various edges, such as 1-2, 3-5 in graph X, and A-B, C-E in graph Y, as indicated. The number of formed edges far exceeds the number of nodes, making node matching a linear assignment problem, while graph matching poses a quadratic assignment problem. Aligning the matched graphs allows the identification of the most similar parts.

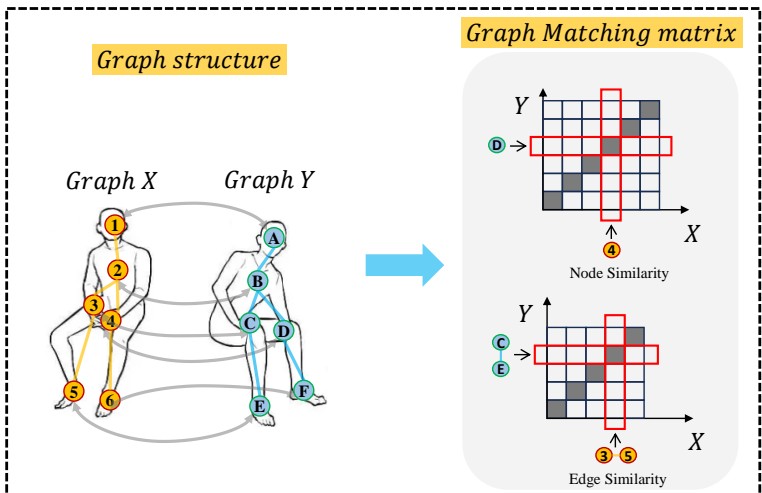

*Figure 5.* The illustration of graph matching. The two graphs to be matched, Graph X and Graph Y, are depicted on the left figure, each annotated with corresponding nodes and partial connections. The diagrams on the right represent the similarity matrices between nodes and between edges.

## A.3. Adaptive algorithm

The specific calculation formula for Sinkhorn is as follows:

$$\mathbf{P} = \exp(\mathbf{M}/\tau),$$
$$P_{ij} \leftarrow \frac{P_{ij}}{\sum_j P_{ij}} \qquad \text{(the sum of each row is 1)},$$
$$P_{ij} \leftarrow \frac{P_{ij}}{\sum_i P_{ij}} \qquad \text{(the sum of each column is 1)}. \quad (7)$$

The second and third lines of the formula represent the process of enforcing bilateral constraints. The second line scales each row to 1, while the third line scales each column to 1.

The delineation of the specific computational process is articulated in Algorithm 2. The diminished complexity results in a notable reduction in computational requirements. Exploiting this attribute makes it particularly apt for mitigating the heightened computational complexity that arises from sparsity or the expansive nature of the assignment matrix.

For the similarity matrices corresponding to two channels, denoted by $R$, in the shallow layers of the neural network, we use the original $R$ for the computation of the permutation matrix. Conversely, in the deep layers of the deep network, we employ the inverse of $R$, which is $-R$, for the computation of the permutation matrix.

Through these two phases, we enable the extraction of richer information in the shallow layers upon task arrival, while facilitating the divergence of different tasks in the deeper layers. This mechanism guarantees the preservation of task distinctiveness by permitting them to traverse separate pathways.

In this paper, we employ the Sinkhorn algorithm; however, when $\tau \leq \tau_{min}$, the Sinkhorn algorithm and the Hungarian algorithm (Kuhn, 1955) exhibit consistent trends.

The Earth Mover's Distance (EMD) algorithm (Rubner et al., 2000) involves solving an optimization problem known as the transportation problem. It is the manifestation of the Sinkhorn algorithm in a low-dimensional space, and the specific algorithmic formula is as follows: Given two probability distributions $P$ and $Q$, represented by histograms $p_i$ and $q_j$ for $i = 1, \ldots, m$ and $j = 1, \ldots, n$ respectively, the EMD can be calculated as follows:

$$\text{EMD}(P, Q) = \min_{\gamma \in \Gamma(p,q)} \sum_{i=1}^{m} \sum_{j=1}^{n} \gamma_{ij} \cdot d(c_i, d_j),$$

where $\Gamma(p, q)$ is the set of all possible transportation plans (joint distributions) between $P$ and $Q$. $\gamma_{ij}$ represents the amount of mass to be transported from $p_i$ to $q_j$. $d(c_i, d_j)$ is the ground distance between the bin $i$ in the source histogram and the bin $j$ in the target histogram. The EMD can be calculated using linear programming techniques:

$$\text{EMD}(P, Q) = \min_{\gamma} \sum_{i=1}^{m} \sum_{j=1}^{n} \gamma_{ij} \cdot d(c_i, d_j),$$

$$\text{s.t.} \quad \sum_{j=1}^{n} \gamma_{ij} = p_i \quad \forall i \in [1, m], \quad \sum_{i=1}^{m} \gamma_{ij} = q_j \quad \forall j \in [1, n], \gamma_{ij} \geq 0 \quad \forall i \in [1, m], \ j \in [1, n].$$

### A.4. The overall framework of `LwI`

The overarching framework of our algorithm operates in Algorithm 3: as tasks stream into the deep network, the new model undergoes training with the input data. Upon completion of training, a fusion of models occurs through maximizing similarity matching in the shallow layers and minimizing similarity matching in the deeper layers. We have observed that our method excels in merging old and new models under data-free conditions, achieving superior task preservation across different tasks. Additionally, our approach, employing misaligned fusion, provides distinct channels for different tasks, better preserving the overall integrity of the deep network.

### A.5. Analysis of time complexity

In the context of our hierarchical matching, the analysis of its time complexity is presented below. Assuming a deep network with $N_L$ layers, each layer containing C channels, the conventional graph matching incurs a time complexity of $O(N^4)$, where N represents the total number of nodes in the graph. However, by adopting a hierarchical matching strategy for deep networks, we can compute the time complexity for each layer individually and subsequently sum them up. As a result, our final time complexity is $O(\frac{1}{N_L^3} N^4)$ determined by this summation:

$$O(\sum_{1}^{N_L} C^4) = O(\sum_{1}^{N_L} (\frac{N}{N_L})^4) = O(\frac{1}{N_L^3} N^4). \tag{8}$$

---

**Algorithm 2** Adaptive algorithm

---

**Input:** Similarity Matrix $R$, Total number of iterations $E$, Parameter $\tau$ for control the difference between Hungarian algorithm and Sinkhorn algorithm ;

**for** *each round* $e = 1, ..., E$ **do**

    **if** $P_i$ *not converged* **then**

        **if** $\tau <= \tau_{min}$ **then**

            |   adaptive algorithm $\leftarrow$ Hungarian algorithm;

        **end**

        **else**

            |   adaptive algorithm $\leftarrow$ Sinkhorn algorithm;

        **end**

        **if** *layer is deep* **then**

            |   $\boldsymbol{R} = -\boldsymbol{R}$

        **end**

        **else**

            |   $\boldsymbol{R} = \boldsymbol{R}$

        **end**

        $\boldsymbol{P} = $ adaptive algorithm$(\boldsymbol{R}, \tau)$;

    **end**

**end**

**Output:** the learned permutation metrics $P_i$.

---

## B. Implementation Details

### B.1. Evaluation

In this paper, we employ two measures, task-agnostic and task-aware, to simultaneously evaluate the performance of these methods in scenarios of known tasks (such as task incremental learning) and unknown tasks (such as class incremental learning). Task-agnostic refers to appending all of the classifier's head to a given data and then taking the maximum value, where the label corresponding to the maximum value is assigned to the category. Task-aware, on the other hand, involves already knowing the task associated with a given data and directly obtaining the maximum value from the corresponding classification head, where the label corresponding to the maximum value is assigned to the data's category. Due to the lack of uniformity in the output of the classification head in our framework, the final performance of Task-agnostic is generally lower than that of Task-aware. Based on the findings in Table 1 and the results below, it is evident that our approach has outperformed even the exemplar-based methods iCaRL and LUCIR in the majority of task-agnostic scenarios.

Assuming that learning has been conducted for $T$ tasks, the model possesses $T$ classification heads corresponding to the tasks indicated as $1$ to $T$, with each classification head containing the respective classes denoted as $n_1, ..., n_T$. Consequently, for the two measurement methodologies mentioned above, we evaluate performance using the following formulas:

$$\text{Accuracy} = \frac{\sum_{k=1}^{N} y_k}{N}, y_k = \begin{cases} 1 & Predict_k == label_k. \\ 0 & else. \end{cases} \tag{9}$$

The formula for the **task-agnostic** method can be expressed as follows: the classification involves selecting the prediction with the highest value from a total of $n_1 + \ldots + n_T$ classes to serve as the final output:

$$Predict_k = argmax([o_0, ..., o_{(n_1+...+n_T-1)}]). \tag{10}$$

The formula for the **task-aware** method is as follows: given that it is the f-th task, the classification involves selecting the prediction with the highest value from a total of $n_f$ classes to serve as the final output:

$$Predict_k = argmax([o_0, ..., o_{(n_f-1)}]). \tag{11}$$

where $o_i$ represents i-th output of the deep network.

In the coarser granularity layers of the neural network, we match the channels with high similarity to enhance mutual common features. Conversely, in the finer granularity layers, we employ minimization of similarity matching to enable misalignment fusion of distinct task features, thereby achieving a protective effect.

---

**Algorithm 3** `LwI`

---

**Input:** Sequential tasks $T_1, ..., T_N$, Sequential data $\{X_1, Y_1\}, \{X_2, Y_2\}, ..., \{X_N, Y_N\}$, New Model for training $model\_new$, Old model for fusion $model\_old$.

Randomly initialize $model\_old$ and $model\_new$

**for** $task\ t = T_1, ..., T_N$ **do**

    /* The training process of $model\_new$                                                             */

    **for** $epoch\ i = 1, ..., n$ **do**

        Initialize $Total\_loss = 0$;

        **if** $t == 1$ **then**

            Update $model\_new\ \boldsymbol{w}_{new}^i \leftarrow \tilde{\boldsymbol{w}}_{new}^i$;

            model training: minimize loss function defined as $\mathcal{L}_{total} = \mathcal{L}_{ce}$;

        **end**

        **else**

            initialize $model\_new\ \boldsymbol{W}_{new} \leftarrow \boldsymbol{W}_{fusion}$;

            Update $model\_new\ \boldsymbol{w}_{new}^i \leftarrow \tilde{\boldsymbol{w}}_{new}^i$;

            model training: minimize loss function defined as $\mathcal{L}_{total} = \mathcal{L}_{ce} + \lambda * \mathcal{L}_{kd}$;

        **end**

    **end**

    /* The training process of $model\_old$                                                                */

    **if** $t == 1$ **then**

        initialize $model\_old\ \boldsymbol{W}_{old} \leftarrow \boldsymbol{W}_{new}$;

    **end**

    **else**

        get $model\_old$ according to Algorithm 1;

    **end**

**end**

---

## B.2. Experiments details

We now validate our method on several benchmark datasets against relevant continual learning baselines. We followed similar experimental setups and framework described in (Masana et al., 2022). We utilized the SGD optimizer for training, and batch sizes for the training, validation and testing sets were consistently set to 64 in all experiments. During network training, the learning rate was initialized at 0.1. Furthermore, the learning rate was decreased by a factor of 0.1 in the 80th and 120th epochs, and the total number of training epochs was set to 200. The model architecture and training hyperparameters are the same for different methods. When employing ResNet32, the momentum for the SGD optimizer was set to 0.9, while, for ResNet18, the momentum for SGD optimizer was set to 0.0.

To gauge the distributional disparity between the new and old models, we introduce divergence as a measurement, and derive the objective of knowledge distillation through the following theoretical deductions:

$$
\begin{aligned}
D_{KL}(p\|q) &= E_{x \sim p(x)} \left( \log \frac{p(x)}{q(x)} \right) \\
&= \sum_{i}^{n} p(x_i) \cdot [\log p(x_i) - \log q(x_i)] \\
&= \sum_{i}^{n} [-p(x_i) \log q(x_i) - (-p(x_i) \cdot \log p(x_i))].
\end{aligned}
\tag{12}
$$

In order to measure the distribution difference between the new and old models, we introduce Kullback-Leibler(KL) Divergence to measure, and get the optimal object of knowledge distillation through the theoretical equation. The last term in Eq.(12)'s final line represents cross-entropy, while the subsequent term signifies entropy. Consequently, when dealing with the same dataset, entropy remains constant, and the divergence between two distributions is determined by cross-entropy.

## B.3. Architecture details

**ResNet32**: The model utilized three convolution blocks, each block containing five convolution layers. The number of output channels ranged from 16 to 32 and culminated in 64. In addition, a fully connected (FC) layer consisting of 64 units was employed, and the output was divided into multiple heads based on task requirements.

**ResNet18**: The model utilized four convolution blocks, with each block containing two convolution layers. The number of output channels ranged from 64 to 128, 256 and increased to 512. A single fully connected (FC) layer with 512 units was employed, and the output was divided into multiple heads based on the task requirements.

*Table 6.* Comparison between different architecture of models.

| Architecture | Total parameters | Model size |
|---|---|---|
| ResNet32 | 466,896 | 1.84MB |
| ResNet18 | 11,220,132 | 42.87MB |

## B.4. Datasets splits details

CIFAR-100 dataset contains 100 classes, each of which contains 600 32*32 color pictures, 500 are for training, and 100 are for testing. The Tiny-Imagenet dataset contains 200 classes, each of which contains 500 64*64 color images, 400 images among which were used for training, 50 used for validation, and 50 for testing.

**CIFAR-100**: If set to 5 splits, it corresponds to 20 classes per head. If set to 10 splits, it corresponds to 10 classes per category. If set to 20 splits, it corresponds to 5 classes per category.

**Tiny-Imagenet**: If set to 5 splits, it corresponds to 40 classes per head. If set to 10 splits, it corresponds to 20 classes per category. If set to 20 splits, it corresponds to 10 classes per category.

## B.5. Baselines

We compared our method with three regularization-based methods, one architecture-based method, and two exemplar-based methods. Regularization-based methods involve adding regularization terms to the loss function during training to protect knowledge from previous tasks. The architecture-based method, specifically the WSN method used in this paper, identifies the optimal subnetwork using masking to achieve continual learning, making it effective under task-aware conditions. Exemplar-based methods involve saving some data from previous tasks, mixing it with the current task's dataset for training, which contributes to uniformity across different task heads and is beneficial for continuous learning in task-agnostic scenarios.

**EWC** (Kirkpatrick et al., 2017): This is a regularization method aimed at protecting previously learned knowledge to prevent forgetting of prior tasks during new task training. It uses the Bayesian formula to constrain the distribution of model parameters, making the crucial parameters from prior tasks less susceptible to modification during new task learning. The formula is shown below:

$$L(\theta) = L_{\text{new}}(\theta) + \lambda \sum_i \frac{1}{2} \Omega_i (\theta_i - \theta_i^*)^2,$$

where $\Omega_i$ respresents the fisher information matrix about parameters.

**SI**: The Path Integral method (SI) (Zenke et al., 2017) accumulates changes in each parameter along the entire learning trajectory in an online manner. The authors of this paper posit that batch updates to weights during parameter updates may lead to an overestimation of importance, while commencing from a pre-trained model may result in its underestimation.

**MAS**: Memory aware synapses(MAS) (Aljundi et al., 2018) computes the regularization term online by accumulating the sensitivity (gradient magnitude) of the learning function.

**RWalk** (Chaudhry et al., 2018): This method integrates the approximation of the Fisher information matrix and online path integral into a single algorithm to compute the importance of each parameter. As the outcomes of this method typically surpass those of SI and MAS methods, the comparative experiments in the main body of this paper employ this approach for evaluation.

**LwF** (Li & Hoiem, 2017): The core concept is to retain the knowledge from previous tasks when learning a new task,

ensuring that the model does not entirely forget the content it has already learned. By employing knowledge distillation, the outputs are aligned to achieve the effect of knowledge preservation.

**WSN** (Kang et al., 2022a): The Lottery Ticket Hypothesis theory is employed, which posits that there exists an optimal path within a neural network for a given task, and this is utilized to apply channel masking. Therefore, this method is typically utilized for tasks with known training and testing processes.

**iCaRL** (Rebuffi et al., 2017): The model incorporates exemplars and employs knowledge distillation to preserve knowledge. The formula is shown below:

$$\ell(\Theta) = - \sum_{(x_i,y_i)\in\mathcal{D}} \left[ \sum_{y=s}^{t} \delta_{y=y_i} \log g_y(x_i) + \delta_{y\neq y_i} \log(1 - g_y(x_i)) \right.$$
$$\left. + \sum_{y=1}^{s-1} q_i^y \log g_y(x_i) + (1 - q_i^y) \log(1 - g_y(x_i)) \right].$$

**LUCIR** (Hou et al., 2019): The use of exemplars is accompanied by the application of several strategies to mitigate the issue of the new class weight vector being larger than the old class, leading to catastrophic forgetting and the model's tendency to classify old class data as new class. In this study, we employed Cosine Normalization, Less-Forget Constraint, and Inter-Class Separation as several methods to alleviate this issue.

Some work has explored the application of pruning methods in continual learning. However, such methods tend to disrupt the overall deep network architecture. Non-structured pruning, in particular, can sometimes lead to more severe consequences. Based on the analysis and experiments mentioned above, we opted to employ a method called "maximizing similarity matching" in the coarser granularity section. This method facilitates the fusion of different deep networks as different tasks occupy denser channels that contain more common features. In the finer granularity section, we employ a method called "minimizing similarity matching" to perform a misalignment fusion of different deep network channels, thereby safeguarding the distinct characteristics of different tasks.

## C. More experiments

### C.1. The performance of various methods when the number of tasks increases by an order of magnitude.

We increased the number of tasks by an order of magnitude for testing, dividing the Tiny-ImageNet dataset into 100 tasks, each with two categories. The comparison methods primarily focus on the latest approaches to parameter protection and the experimental results are shown in Figure 6. From the Figure, we can observe that our method achieves the best performance when compared to other approaches, The analysis suggests that the use of isolation-based methods reduces the number of learnable parameters in the network, leading to decreased learning ability for subsequent tasks and demonstrates that our pathway protection approach can preserve knowledge of old tasks while generalizing to new task knowledge.

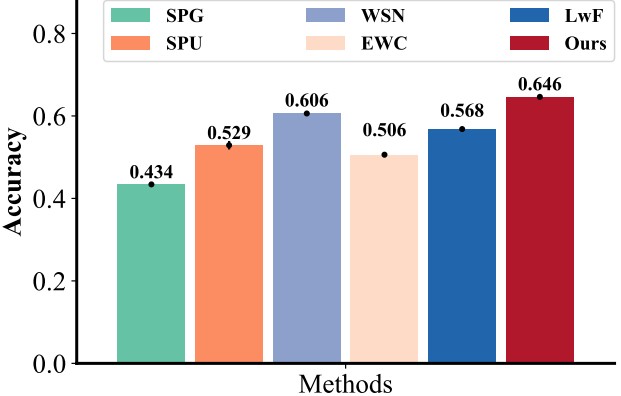

*Figure 6.* Task-aware accuracy of methods when the number of tasks is 100.

**C.2. Discussion about Table 1**

Through the comparison of results and analysis of the four experimental sets, we summarize the findings and elucidate the underlying reasons.

Firstly, regarding the network model capacity, we posit that, under identical scenarios, the gradual increase in model capacity leads to a sparser channel occupancy. This sparsity constitutes a key aspect of our proposed methodology. Thus, the conclusions drawn from the experiments, particularly the higher performance gains achieved by ResNet18 over ResNet32 under task-agnostic conditions, validate the correctness of our proposed task diversification concept, as depicted in Figure 2.

Secondly, with respect to the dataset, our observations indicate that under equivalent deep network architectures, the superiority of our method becomes more pronounced with increasing dataset complexity. This emphasizes the efficacy of our approach in handling intricate datasets. EWC and RWalk methods are designed to address issues arising from significant data variations, making it challenging for these regularizations to effectively constrain parameter shifts. LwF, primarily employed for training different tasks, experiences the blending of task knowledge, as illustrated in Figure 2. This blending is likely to result in outcomes inferior to our method. WSN requires a mask when dealing with various tasks, limiting its applicability to task-aware testing. Additionally, as the number of tasks increases, the reduction in learnable parameters diminishes its effectiveness. iCaRL and LUCIR methods benefit from partial datasets of all previous tasks during the training of subsequent tasks, offering advantages for task-agnostic testing.

Thirdly, in the situation of task agnostic, our deep network exhibits lower performance compared to exemplar-based continual learning (CL) methods on the CIFAR-100 dataset. We posit that, while our deep network learns each task individually, the persistent setup of learning classification heads results in inconsistent output sizes for these task-specific heads, thereby posing challenges in scenarios of task uncertainty. The utilization of exemplars involves incorporating partial data from previous tasks into the current dataset during training, mitigating the inconsistency in classification heads. However, with the complexity of datasets such as Tiny-Imagenet, the performance improvement derived from exemplar usage is surpassed by the benefits brought about by our approach of task-specific streams.

**C.3. Experiments analysis in Table 1**

According to the experiment on dataset CIFAR-100, architecture ResNet32, our approach surpasses the baseline performance of all non-replay pools in the comparative experiments. Compared to the best-performing regularization method LwF, our approach demonstrates a maximum improvement of 5.88% under task-agnostic conditions. In scenarios of task awareness, the performance is further enhanced, showing an improvement of 6.28%. In comparison to the WSN method, which primarily designed for task incremental learning, hence not applicable to scenarios of task agnosticism. Under the task-aware setting, our method achieves an approximately 1% improvement. When contrasted with exemplar-based approaches, our method achieves its peak performance under task-aware 5/10 splits conditions.

With the increase in model size of deep network, the improvement of our method becomes more pronounced under task-agnostic conditions. As shown in the second block, that is the experiment on dataset CIFAR-100, architecture ResNet18, our method outperforms other comparative approaches under task-aware conditions. In all other conditions, our method surpasses the performance of the methods employed in the comparative experiments. When compared to the best-performing regularization method, LwF, our approach exhibits a maximum improvement of 5.04% under task-aware conditions and an even more substantial improvement of 7.29% under task-agnostic conditions. In contrast to the WSN method, our approach demonstrates a performance improvement of around 2.45% in task-aware scenarios. In comparison with exemplar-based approaches, our method attains its peak performance under task-aware 10/20 splits conditions.

**C.4. Experiments on Tiny-Imagenet dataset using ResNet32.**

According to Table 7 and 8, our method surpasses almost all comparative results, except for iCaRL under task-agnostic 5-splits conditions. In comparison to the LwF method, our approach exhibits a maximum improvement of up to 3.1% under task-agnostic conditions and 7.18% under task-aware conditions. When contrasted with the LUCIR method, our performance surpasses by a maximum of 3.15% under task-agnostic conditions and 10.25% under task-aware conditions.

Under conditions of task-agnostic, analysis of the results in Table 7 reveals that our method, with the exception of a slight underperformance compared to the iCaRL method in the 5-splits scenario, consistently outperforms the comparative experiments in all other cases. In comparison to the LwF method, which exhibits the best performance among regularization

methods, our approach demonstrates an improvement of up to 3.1%. Furthermore, when contrasted with the EWC method, our method achieves a maximum improvement of 9.61%. Notably, when compared to exemplar-based methods on the Tiny-Imagenet dataset, our approach even surpasses them, highlighting the advantages of our task-shifting methodology. This is evident in the ability of our method to achieve higher activation levels for each channel corresponding to a specific task, even without the unification of classification heads. Thus, task specialization is achieved, with the activation intensity for each channel surpassing that of all other tasks, emphasizing the effectiveness of our task-shifting approach.

*Table 7.* Task-agnostic accuracy (%) of methods on the Tiny-Imagenet dataset based on the architecture of ResNet32.

| Method | Exemplar | Task-agnostic | | |
|---|---|---|---|---|
| | | 5 splits | 10 splits | 20 splits |
| EWC | no | $7.76 \pm 0.75$ | $3.80 \pm 0.32$ | $2.60 \pm 0.19$ |
| RWalk | | $11.10 \pm 0.35$ | $4.71 \pm 0.21$ | $4.54 \pm 0.63$ |
| LwF | | $20.12 \pm 0.63$ | $13.72 \pm 0.52$ | $9.11 \pm 0.40$ |
| **Ours** | | $\mathbf{22.21 \pm 0.39}$ | $\mathbf{16.75 \pm 0.21}$ | $\mathbf{12.21 \pm 0.29}$ |
| iCaRL | 2000 | $\mathbf{22.45 \pm 0.14}$ | $16.48 \pm 0.84$ | $9.94 \pm 0.24$ |
| LUCIR | | $20.05 \pm 0.16$ | $13.60 \pm 0.42$ | $10.38 \pm 0.40$ |

Under conditions of task-aware, examination of the results in Table 8 reveals that our method consistently outperforms the comparative experiments.

In comparison to the LwF method, which demonstrates the best performance among regularization methods, our approach exhibits a maximum improvement of up to 7.18%. Furthermore, when contrasted with the RWalk method, our approach surpasses it even more significantly, reaching up to 25.71%. In comparison to the architecture-based method WSN, the advantages of our method under conditions of task knowledge are not particularly pronounced, with the highest improvement being 2.36%. However, when compared to exemplar-based methods, the superiority of our approach becomes notably evident.

*Table 8.* Task-aware accuracy (%) of methods on the Tiny-Imagenet dataset based on the architecture of ResNet32.

| Method | Exemplar | Task-aware | | |
|---|---|---|---|---|
| | | 5 splits | 10 splits | 20 splits |
| EWC | no | $31.97 \pm 1.18$ | $36.71 \pm 1.23$ | $42.91 \pm 0.31$ |
| RWalk | | $43.75 \pm 2.08$ | $43.22 \pm 2.11$ | $37.66 \pm 1.74$ |
| LwF | | $47.78 \pm 0.98$ | $52.39 \pm 0.60$ | $56.19 \pm 0.54$ |
| WSN | | $50.06 \pm 0.37$ | $56.29 \pm 0.75$ | $63.24 \pm 0.87$ |
| **Ours** | | $\mathbf{52.42 \pm 0.59}$ | $\mathbf{57.84 \pm 1.15}$ | $\mathbf{63.37 \pm 0.44}$ |
| iCaRL | 2000 | $44.87 \pm 0.61$ | $49.42 \pm 1.29$ | $54.43 \pm 0.73$ |
| LUCIR | | $44.53 \pm 0.68$ | $46.86 \pm 1.29$ | $53.12 \pm 0.73$ |

Based on the above experimental results, we observe that our method not only accomplishes task channel specialization under conditions of task agnostic without the need for deliberate unification of classification heads but also, under conditions of task awareness, exhibits a comparative advantage. In contrast to other methods, our approach shows minimal catastrophic forgetting of previously acquired knowledge and, in certain instances, even demonstrates a facilitating effect. This observed promotion of cooperation among tasks is a notable outcome of our method.

### C.5. Experiments on CIFAR-100 dataset using AlexNet.

According to Table 9 and 10, our method surpasses all comparative results.

Under conditions of task-agnostic, analysis of the results in Table 9 reveals that our method consistently outperforms the comparative experiments in all cases. In comparison to the LwF method, which exhibits the best performance among regularization methods, our approach demonstrates an improvement of up to 2.0%. Furthermore, when contrasted with the EWC method, our method achieves a maximum improvement of 16.1%. Thus, this is evident that task specialization is achieved, with the activation intensity for each channel surpassing that of all other tasks, emphasizing the effectiveness of our task-shifting approach.

Under conditions of task-aware, examination of the results in Table 10 reveals that our method consistently outperforms the comparative experiments. In comparison to the LwF method, which demonstrates the best performance among regularization methods, our approach exhibits a maximum improvement of up to 1.8%. Furthermore, when contrasted with the SI method,

*Table 9.* Task-agnostic accuracy (%) of methods on the CIFAR-100 dataset based on the architecture of AlexNet.

| Method | Exemplar | Task-agnostic | | |
|---|---|---|---|---|
| | | 5 splits | 10 splits | 20 splits |
| EWC | | $13.8 \pm 1.4$ | $6.9 \pm 2.0$ | $4.5 \pm 0.8$ |
| SI | | $14.2 \pm 1.4$ | $6.8 \pm 2.0$ | $3.8 \pm 0.3$ |
| RWalk | no | $14.0 \pm 1.7$ | $8.4 \pm 1.4$ | $3.7 \pm 1.4$ |
| MAS | | $14.3 \pm 1.1$ | $8.2 \pm 0.9$ | $5.3 \pm 0.9$ |
| LwF | | $27.9 \pm 1.7$ | $19.5 \pm 1.6$ | $10.7 \pm 1.1$ |
| **Ours** | | $\mathbf{29.9 \pm 0.6}$ | $\mathbf{20.4 \pm 0.9}$ | $\mathbf{11.2 \pm 1.1}$ |

our approach surpasses it even more significantly, reaching up to 27.8%. In comparison to EWC method, the advantages of our method, with the highest improvement being 26.6%.

*Table 10.* Task-aware accuracy (%) of methods on the CIFAR-100 dataset based on the architecture of AlexNet.

| Method | Exemplar | Task-aware | | |
|---|---|---|---|---|
| | | 5 splits | 10 splits | 20 splits |
| EWC | | $34.6 \pm 2.0$ | $38.9 \pm 2.7$ | $45.5 \pm 3.2$ |
| SI | | $35.9 \pm 1.2$ | $37.7 \pm 1.2$ | $43.7 \pm 3.2$ |
| RWalk | no | $37.2 \pm 1.7$ | $38.5 \pm 1.1$ | $43.9 \pm 2.9$ |
| MAS | | $37.1 \pm 1.3$ | $42.1 \pm 1.9$ | $50.5 \pm 4.0$ |
| LwF | | $58.8 \pm 1.1$ | $64.8 \pm 1.8$ | $68.6 \pm 0.8$ |
| **Ours** | | $\mathbf{60.6 \pm 0.5}$ | $\mathbf{65.5 \pm 0.8}$ | $\mathbf{68.8 \pm 1.0}$ |

Based on the above experimental results, we observe that our method not only accomplishes task channel specialization under conditions of task agnostic without the need for deliberate unification of classification heads but also, under conditions of task awareness, exhibits a comparative advantage. In contrast to other methods, our approach shows minimal catastrophic forgetting of previously acquired knowledge and, in certain instances, even demonstrates a facilitating effect. This observed promotion of cooperation among tasks is a notable outcome of our method.

### C.6. Experiments on different similarity measurement formulas.

In this study, the Euclidean distance and cosine similarity were employed to measure the distance between two model channels.

**The Euclidean distance** primarily quantifies the distance between two vectors in space, with smaller absolute values indicating closer proximity. It is a commonly used distance measurement formula. On the other hand,

**Cosine similarity** gauges the angle between two vectors within the same sphere, mainly reflecting directional differences. Larger numerical values denote smaller angle discrepancies, indicating closer proximity in space. It is a widely used formula for measuring similarity.

This section compared and validated the use of Euclidean distance and cosine similarity to measure channel proximity and revealed that, in most cases, using distance measurement is preferable to using cosine similarity.

$$\text{Euclidean Distance} = \|a - b\|_2 \tag{13}$$

$$\text{Cosine Similarity} = \frac{a \cdot b}{\|a\| \cdot \|b\|} \tag{14}$$

where $a$ and $b$ represent two vectors.

According to Table 11 and 12, it is observed that under two different testing conditions, when measuring model similarity for the purpose of model fusion, the use of Euclidean distance consistently yields slightly higher performance compared to cosine similarity. This trend holds true across various scenarios, with the notable exception of the task-agnostic 10-splits condition, where results obtained using Euclidean distance are recorded at 30.62%, while those using cosine similarity are slightly higher at 31.16%. Consequently, based on the comparative experimental outcomes presented in this paper, the choice is made to employ Euclidean distance for model fusion, facilitating a comprehensive evaluation of testing effectiveness.

*Table 11.* Task-agnostic accuracy (%) of methods between different similarity measurement formulas.

| Dataset | Architecture | Method | Task-agnostic | | |
|---|---|---|---|---|---|
| | | | 5 splits | 10 splits | 20 splits |
| CIFAR-100 | ResNet32 | **Ours** | **43.42 ± 0.58** | 30.62 ± 1.08 | **20.31 ± 0.77** |
| | | Ours with cosine | 43.09 ± 0.95 | **31.16 ± 0.78** | 20.03 ± 0.93 |
| CIFAR-100 | ResNet18 | **Ours** | **51.95 ± 0.56** | 36.36 ± 1.06 | **22.99 ± 0.39** |
| | | Ours with cosine | 51.78 ± 0.92 | **37.06 ± 1.12** | 22.77 ± 0.57 |
| Tiny-Imagenet | ResNet32 | **Ours** | 22.21 ± 0.39 | 16.75 ± 0.21 | **12.21 ± 0.29** |
| | | Ours with cosine | **22.26 ± 0.60** | **16.96 ± 0.16** | 11.94 ± 0.41 |

*Table 12.* Task-aware accuracy (%) of methods between different similarity measurement formulas.

| Dataset | Architecture | Method | Task-aware | | |
|---|---|---|---|---|---|
| | | | 5 splits | 10 splits | 20 splits |
| CIFAR-100 | ResNet32 | **Ours** | **76.10 ± 0.33** | **81.12 ± 0.90** | **83.19 ± 0.35** |
| | | Ours with cosine | 75.63 ± 0.46 | 79.35 ± 0.86 | 83.05 ± 0.36 |
| CIFAR-100 | ResNet18 | **Ours** | **81.10 ± 0.80** | **84.90 ± 0.36** | **86.49 ± 0.55** |
| | | Ours with cosine | 80.72 ± 1.09 | 84.01 ± 1.02 | 85.91 ± 0.85 |
| Tiny-Imagenet | ResNet32 | **Ours** | 52.42 ± 0.59 | 57.84 ± 1.15 | **63.37 ± 0.44** |
| | | Ours with cosine | **52.68 ± 0.53** | **58.37 ± 0.90** | 62.59 ± 0.81 |

## C.7. Experiments on without using knowledge distillation module.

According to Table 13, in order to verify the effectiveness of our method, we also carried out ablation experiments on the knowledge distillation module, and the results showed that in this case, the knowledge generation would be shifted to a large extent, thus reducing the effect.

*Table 13.* Task-aware accuracy (%) of our methods without using knowledge distillation module.

| Dataset | Architecture | Method | Task-aware | | |
|---|---|---|---|---|---|
| | | | 5 splits | 10 splits | 20 splits |
| CIFAR-100 | ResNet18 | **Ours** | **81.10 ± 0.80** | **84.90 ± 0.36** | **86.49 ± 0.55** |
| | | Ours w/o KD | 75.91 ± 0.85 | 73.91 ± 0.51 | 71.98 ± 1.24 |

## C.8. An example of whether to use knowledge distillation module.

According to Table 14, Table 15 and Table 16, applying knowledge distillation to each layer method results in minimal changes in the model's parameter space. Conversely, without using knowledge distillation method leads to significant differences. The following three tables depict the accuracy(%) obtained from utilizing the ResNet32 model under the same conditions for five tasks on the CIFAR-100 dataset. It can be observed that our method sometimes achieves better performance after training on new tasks than after the initial training.

*Table 14.* Task-aware accuracy (%) of our method using knowledge distillation module for every layer.

| Task-ID | Task1 | Task2 | Task3 | Task4 | Task5 | Overall |
|---|---|---|---|---|---|---|
| Task1 | 78.2 | 0 | 0 | 0 | 0 | 78.2 |
| Task2 | 78.0 | 68.1 | 0 | 0 | 0 | 73.1 |
| Task3 | 77.6 | 63.1 | 68.7 | 0 | 0 | 69.8 |
| Task4 | 74.4 | 59.8 | 62.0 | 64.7 | 0 | 65.2 |
| Task5 | 74.2 | 60.9 | 53.5 | 65.0 | 63.0 | 63.3 |

*Table 15.* Task-aware accuracy (%) of our method.

| Task-ID | Task1 | Task2 | Task3 | Task4 | Task5 | Overall |
|---------|-------|-------|-------|-------|-------|---------|
| Task1 | 78.2 | 0 | 0 | 0 | 0 | 78.2 |
| Task2 | 75.3 | 74.2 | 0 | 0 | 0 | 74.8 |
| Task3 | 74.8 | 76.6 | 75.2 | 0 | 0 | 75.5 |
| Task4 | 75.3 | 76.3 | 75.7 | 76.0 | 0 | 75.8 |
| Task5 | 74.2 | 75.3 | 75.5 | 77.2 | 76.1 | 75.7 |

*Table 16.* Task-aware accuracy (%) of our method without using knowledge distillation module.

| Task-ID | Task1 | Task2 | Task3 | Task4 | Task5 | Overall |
|---------|-------|-------|-------|-------|-------|---------|
| Task1 | 78.2 | 0 | 0 | 0 | 0 | 78.2 |
| Task2 | 70.1 | 77.7 | 0 | 0 | 0 | 73.9 |
| Task3 | 61.0 | 73.0 | 72.7 | 0 | 0 | 68.9 |
| Task4 | 54.8 | 69.0 | 65.8 | 84.6 | 0 | 68.5 |
| Task5 | 52.3 | 59.5 | 58.9 | 78.4 | 81.7 | 66.2 |

## C.9. Experiments on the validation of effectiveness with task diversion module.

In order to validate the effectiveness of our proposed task diversion method, we compared it with an approach that does not perform deep-level minimization of similarity. This approach involves using maximization of similarity at all layers for model fusion. Our findings indicate that our method yields better results as the complexity of the task increases.

The entry labeled **"Ours w/o task diversion"** in the Table 17 and 18 signifies that each layer of the deep network model employs maximization of similarity matching, signifying the absence of task-specific parameter diversion during model fusion. As evidenced by the results, our approach incorporates the minimization of similarity matching in the final layer, facilitating channel diversion for task segregation and consequently ensuring protection across distinct tasks. Consequently, under equivalent conditions, our method consistently outperforms approaches solely relying on matching the channels with high similarity.

*Table 17.* Task-agnostic accuracy (%) of methods on the validation of effectiveness with task diversion module.

| Dataset | Architecture | Method | Task-agnostic | | |
|---------|--------------|--------|----------|-----------|-----------|
| | | | 5 splits | 10 splits | 20 splits |
| CIFAR-100 | ResNet32 | **Ours** | **43.42 ± 0.58** | **30.62 ± 1.08** | **20.31 ± 0.77** |
| | | Ours w/o task diversion | 41.73 ± 0.41 | 29.94 ± 0.89 | 18.02 ± 0.97 |
| CIFAR-100 | ResNet18 | **Ours** | **51.95 ± 0.56** | **36.36 ± 1.06** | **22.99 ± 0.39** |
| | | Ours w/o task diversion | 51.13 ± 0.43 | 35.70 ± 0.88 | 21.49 ± 0.54 |
| Tiny-Imagenet | ResNet32 | **Ours** | **22.21 ± 0.39** | **16.75 ± 0.21** | **12.21 ± 0.29** |
| | | Ours w/o task diversion | 20.80 ± 0.46 | 15.31 ± 0.75 | 10.68 ± 0.53 |

*Table 18.* Task-aware accuracy (%) of methods on the validation of effectiveness with task diversion module.

| Dataset | Architecture | Method | Task-aware | | |
|---------|--------------|--------|----------|-----------|-----------|
| | | | 5 splits | 10 splits | 20 splits |
| CIFAR-100 | ResNet32 | **Ours** | **76.10 ± 0.33** | **81.12 ± 0.90** | **83.19 ± 0.35** |
| | | Ours w/o task diversion | 75.54 ± 1.37 | 79.38 ± 1.02 | 81.43 ± 1.51 |
| CIFAR-100 | ResNet18 | **Ours** | **81.10 ± 0.80** | **84.90 ± 0.36** | **86.49 ± 0.55** |
| | | Ours w/o task diversion | 80.17 ± 0.35 | 83.60 ± 0.86 | 84.15 ± 0.96 |
| Tiny-Imagenet | ResNet32 | **Ours** | **52.42 ± 0.59** | **57.84 ± 1.15** | **63.37 ± 0.44** |
| | | Ours w/o task diversion | 50.26 ± 0.66 | 55.07 ± 1.18 | 60.98 ± 1.15 |

## C.10. Experiments on using minimum similarity matching on different layers.

In order to determine the most effective layers for performance improvement through minimizing similarity matching, we conducted extensive comparative experiments with the ResNet32 model. Specifically, we tested the impact of the final layer, last two layers, last three layers, and last four layers. Remarkably, we observe very similar results across these configurations.

According to Table 19 and 20, we observe that when minimizing similarity matching for the final two, three, and four layers, the ultimate results are comparable to those obtained by minimizing similarity matching for a single layer. However, in the majority of cases, the performance is lower than when minimizing similarity matching for just one layer. Therefore, based on the results of our previous comparative experiments, we opted to minimize similarity matching for the final layer.

*Table 19.* Task-agnostic accuracy (%) of methods on using minimum similarity matching on different layers.

| Dataset | Architecture | Method | Task-agnostic | | |
| --- | --- | --- | --- | --- | --- |
| | | | 5 splits | 10 splits | 20 splits |
| Tiny-Imagenet | ResNet32 | **Ours** | **$22.21 \pm 0.39$** | **$16.75 \pm 0.21$** | **$12.21 \pm 0.29$** |
| | | Ours 2layers | $22.05 \pm 0.60$ | $16.43 \pm 0.38$ | $11.40 \pm 0.21$ |
| | | Ours 3layers | $21.65 \pm 0.87$ | $16.65 \pm 0.37$ | $11.44 \pm 0.31$ |
| | | Ours 4layers | $21.61 \pm 0.79$ | $16.66 \pm 0.17$ | $11.54 \pm 0.27$ |

*Table 20.* Task-aware accuracy (%) of methods on using minimum similarity matching on different layers.

| Dataset | Architecture | Method | Task-aware | | |
| --- | --- | --- | --- | --- | --- |
| | | | 5 splits | 10 splits | 20 splits |
| Tiny-Imagenet | ResNet32 | **Ours** | **$52.42 \pm 0.59$** | **$57.84 \pm 1.15$** | **$63.37 \pm 0.44$** |
| | | Ours 2layers | $52.41 \pm 0.58$ | $56.10 \pm 0.59$ | $61.01 \pm 0.58$ |
| | | Ours 3layers | $51.80 \pm 0.81$ | $56.60 \pm 0.31$ | $60.85 \pm 1.07$ |
| | | Ours 4layers | $52.20 \pm 0.53$ | $56.84 \pm 0.22$ | $61.33 \pm 0.67$ |

According to Table 19 and 20, we observe that when minimizing similarity matching for the final two, three, and four layers, the ultimate results are comparable to those obtained by minimizing similarity matching for a single layer. However, in the majority of cases, the performance is lower than when minimizing similarity matching for just one layer. Therefore, based on the results of our previous comparative experiments, we opted to minimize similarity matching for the final layer.

## C.11. Devices

In the experiments, we conduct all methods on a local Linux server that has two physical CPU chips (Intel(R) Xeon(R) CPU E5-2640 v4 @ 2.40GHz) and 32 logical kernels. All methods are implemented using Pytorch framework and all models are trained on GeForce RTX 2080 Ti GPUs.

