# OpenReview forum: "Learning without Isolation: Pathway Protection for Continual Learning"
_ICML.cc/2025/Conference — ICML 2025 poster_

### Official Review · Reviewer_bzga · 2025-03-09

**Overall Recommendation:** 4

**Summary:**

The paper introduces a novel continual learning method specifically designed to address the problem of catastrophic forgetting, which is a significant challenge in the field of machine learning when models are required to learn from a sequence of tasks. The proposed approach leverages pathway protection techniques, which aim to preserve the essential parts of the neural network that are crucial for previously learned tasks while allowing the network to adapt to new tasks. Unlike traditional methods that focus on isolating parameters for each task or using memory replay to store past experiences, pathway protection emphasizes maintaining the integrity of critical pathways within the network. This mechanism not only helps to safeguard previously learned representations but also facilitates the efficient integration of new knowledge, enabling the model to continue learning without losing valuable information from prior tasks. By doing so, the proposed method aims to strike a balance between retaining old knowledge and acquiring new skills, ultimately leading to more robust and scalable continual learning systems.

**Claims And Evidence:**

The claims presented in the paper are largely substantiated by comprehensive experimental results, demonstrating the effectiveness of the proposed method for mitigating catastrophic forgetting in continual learning scenarios.

Nevertheless, there appear to be some details missing regarding the use of OT in Section 3.2. OT is typically used to find the minimum effort required to transport one distribution to another. However, in the paper, you use OT, specifically the Sinkhorn approximation, to 'transform a binary 0-1 matrix into a soft matching matrix with a sum of 1 through a process of bi-directional relaxation.' Could you kindly clarify how this approach relates to the traditional use of OT? It would be helpful to elaborate on this process further in the main text.

**Essential References Not Discussed:**

Related works that rely on different paths in a network and sparsity are not cited and compared to qualitatively or empirically, such as [1, 2, 3].

[1] PathNet: https://arxiv.org/abs/1701.08734

[2] DEN: https://arxiv.org/abs/1708.01547

[3] APD: https://arxiv.org/abs/1902.09432

**Experimental Designs Or Analyses:**

1.The authors have selected a wide range of well-established benchmark datasets for their experiments, including CIFAR-10, CIFAR-100, and Tiny-ImageNet, to assess the effectiveness of their proposed method in various continual learning settings.

2.The authors clearly define their experimental setup, including the use of ResNet18 and ResNet32 architectures for evaluating the accurate rates on the given datasets.

3.One of the strengths of the experimental design is the inclusion of a series of ablation studies to evaluate the individual components of the proposed method. These studies help isolate the effects of key factors, such as pathway protection and the graph-based integration of model weights, on the overall performance. By systematically varying the design choices (e.g., whether pathway protection is applied or not), the authors are able to demonstrate the contributions of each component to the method’s success in mitigating catastrophic forgetting.

4.The paper includes experiments in both task-incremental and class-incremental settings, two common setups in continual learning.

**Methods And Evaluation Criteria:**

While the authors have demonstrated the feasibility of the proposed method across various techniques, datasets, and architectures, I have a few suggestions for further improvement.

1.Although the authors evaluated forgetting rates using the ResNet18 architecture on the CIFAR-100 dataset, it would still be beneficial to test the forgetting rates of ResNet32 on CIFAR-100 and ResNet18 on the Tiny-ImageNet dataset.

2.Related works that rely on different paths in a network and sparsity are not cited and compared to qualitatively or empirically [1,2,3].

**Other Comments Or Suggestions:**

Please address the concerns I raised in the weaknesses section. Based on your responses, I would be happy to reconsider my scores.

**Other Strengths And Weaknesses:**

Stengths:

1.	The authors propose a method called Learning without Isolation (LwI), where, at each step in continual learning, a new model is trained on a task while distilling knowledge from an existing model. This idea aligns closely with the concept of information flow in the human brain, while also integrating the characteristics of neural networks into this research.

2.	The authors conceptualize the model’s parameters as a graph, they construct paths as convex combinations of new and old model weights, applying a permutation matrix to the new model's weights. For shallow layers, the permutation matrix is determined by the similarity between adjacent layer weights, while for deeper layers, it is based on the negative similarity between graph nodes. This encourages shared paths across different tasks, promoting feature reuse in the shallow layers, and fosters the use of distinct paths in the deeper layers.

3.	Extensive experiments are conducted on image classification continual learning tasks, utilizing benchmarks such as CIFAR-10, CIFAR-100, and Tiny-ImageNet, as well as using different architectures of neural networks to validate the effectiveness of proposed method. The authors examine both task-agnostic and task-aware settings.

Weaknesses:

In the experiments presented in the paper, the authors evaluate the method on a maximum of 20 tasks. This raises an important question: how does the model perform when the number of tasks is increased by an order of magnitude? It seems reasonable to expect that as the number of tasks grows, there will be a tradeoff between the network's ability to accommodate more tasks and the overall model size. This tradeoff is likely to impact both the efficiency of the network and its capacity to preserve knowledge across tasks.

**Questions For Authors:**

Please check the above concerns.

**Relation To Broader Scientific Literature:**

The proposed method in this paper draws inspiration from concepts found in neuroscience, particularly regarding how the human brain handles learning and memory retention over time. In continual learning, one of the primary challenges is catastrophic forgetting—where the model "forgets" previously learned information as new tasks are learned. This mirrors how human brains can forget previously learned information when new knowledge is acquired, a phenomenon known as interference in cognitive psychology.

Recent research in neuroscience has shown that the brain employs mechanisms such as synaptic consolidation and neuroplasticity to minimize forgetting. In particular, the brain strengthens and modifies synaptic connections between neurons as new information is learned while maintaining previously established pathways. This process ensures that long-term memories are not easily overwritten. The method proposed in this paper mimics this idea by implementing pathway protection strategies, which involve protecting previously learned features or representations during the learning of new tasks, in a manner similar to how the brain "protects" neural connections when learning new information.

**Theoretical Claims:**

The authors have provided a detailed explanation of the theories used.

1.	Pathway Protection Enhances Knowledge Retention (Guaranteed by sparsity)

The theoretical foundation of the proposed method is rooted in the concept of pathway protection, which is designed to safeguard key pathways or network parameters that represent learned knowledge from previous tasks.

2.	Graph Matching Method

The authors propose a graph-based method for integrating weights from previous models into the current model, which theoretically capitalizes on the structural similarity between shallow and deep layers. Shallow layers are assumed to share common representations across tasks, while deeper layers specialize in more task-specific features.

---

> ### Author Rebuttal · Authors · 2025-03-31
>
> **Response to Claims:** Thanks for pointing out the issue. We would like to explain it as follows.
>
> 1. In continual learning, considering the lack of correspondence between neurons in Model 1 and Model 2, it is possible that the function of the p-th neuron in Model 1 is very similar to that of the (p+1)-th neuron in Model 2, despite their different positions. Therefore, when using the OT algorithm in neural networks, **we aim to optimally (minimizing cost) transfer neurons from a specific layer in Model 1 to the corresponding layer in Model 2**, achieving model alignment.
> 2. **The advantages of using a soft matching algorithm.**
>    - Using entropy regularization and bidirectional relaxation ensures that the optimization function is smoother.
>    - **One-to-one matching hinders knowledge sharing.** One weight in Model 1 might be similar to multiple weights in Model 2. In such cases, using one-to-one matching may set the corresponding weights with similar knowledge to zero, hindering knowledge sharing.
> 3. We compared direct fusion and fusion using hard matching, and the results show that soft matching is more advantageous. ResNet18-based model on the CIFAR-100 dataset, measured under task-aware scenarios.
>
> |Method|5 splits|10 splits|20 splits
> |-|-|-|-
> |Ours w/o alignment|66.13±0.98|68.80±1.65|70.30±1.27
> |Ours with OT|75.73±0.58|77.73±0.76|80.60±0.71
> |**Ours**|**81.10±0.80**|**84.90±0.36**|**86.49±0.55**
>
> **Response to Essential References:** Thanks for the meaningful suggestions. We indeed omitted the comparisons in our manuscript.
>
> **Comparison with [1]**
> 1. **The different approaches for protecting task-related knowledge.** [1] employs a genetic algorithm. It selects the parameter subset for the next task while fixing the important parameters of the previous tasks, whereas our method achieves knowledge protection through a matching-based approach.
> 2. [1] does not train frozen parameters, whereas we train all parameters.
>
> **Comparison with [2]**
> 1. [2] selectively retrains network parameters and adapts to different tasks by dynamically changing neurons. However, our method protects knowledge of different tasks by finding the optimal pathways for each task and using matching techniques.
> 2. The method proposed in [2] dynamically increases network capacity. In contrast, our method uses a fixed network capacity approach.
>
> **Comparison with [3]**
> 1. [3] protects knowledge through parameter decomposition and masking techniques. However, our method protects important pathways through matching techniques.
> 2. [3] focuses on protecting network connections between adjacent layers, whereas we consider a complete pathway from input to output.
>
> **Response to Weakness:** Thanks for the valuable question.
>
> - Regarding the number of learnable tasks, our method employs soft protection, allowing for a greater number of learnable tasks. Compared to hard mask methods like SupSup, we use all network parameters and protect important task paths through misaligned fusion. Based on the reviewer's suggestions, we conducted experiments on Tiny-ImageNet with 100 tasks, each comprising two classes. The results show that our method performs very well.
> - To verify the performance of our method when increasing the task number by an order of magnitude, we split Tiny-ImageNet into 100 tasks and conducted tests using different methods. The results show that our method achieves better performance.
>
> | Method | SPG |SPU	|WSN	|EWC	|LwF	|Supsup	| Ours |
> | --- |  ---  | --- |  --- |---| --- |  ---  | --- |
> | 100 splits | 43.40±0.45|	52.90±1.02|	60.60±0.43|	50.61±0.61|	56.82±0.55|	49.47±0.52 | **64.63±0.14**|

---

> > ### Comment · Reviewer_bzga · 2025-04-02
> >
> > The rebuttal emphasizes the strengths of the proposed method in optimal knowledge transfer, effective task protection, and scalability. By leveraging the OT algorithm for neuron alignment and soft matching techniques for smoother optimization, the method ensures better knowledge retention compared to existing approaches. Unlike dynamic neuron adaptation or hard-masking methods, it protects knowledge via a fixed network capacity approach while enabling a complete knowledge flow from input to output. Experiments on CIFAR-100 and Tiny ImageNet (100 tasks) further validate its superior performance and scalability. Based on these points, I have decided to raise my score to 4.

---

> > > ### Author Response · Authors · 2025-04-02
> > >
> > > **Many thanks for raising the score!**
> > >
> > > Thank you very much for your insightful suggestions, which have been greatly enlightening and are crucial for enhancing the quality of our paper! Comparing our approach with provided methods and the latest techniques will enhance the competitiveness of our paper. Additionally, investigating OT methods will contribute to improving the quality of our research. The discussion regarding more detailed issues has further deepened the analysis of the experimental results, which can enhance the persuasiveness of our paper. We will adhere to these suggestions in the final version and also revise the paper according to all other comments.

---

### Official Review · Reviewer_Wbqt · 2025-03-10

**Overall Recommendation:** 4

**Summary:**

The paper proposes a novel continual learning framework that assigns distinct neural pathways to different tasks, enabling knowledge retention while replacing traditional masking & pruning methods. The authors use graph matching for model fusion, leveraging neural network properties by maximizing similarity alignment in shallow layers and minimizing it in deeper layers. Knowledge distillation is applied to constrain parameter deviations. The framework is validated on networks of different sizes and datasets of different sizes.

**Claims And Evidence:**

Most claims made in the submission are clear. However, there are some areas that need further clarification:
1. Although the 'Activation level' in Figure 2 is explained in the text, some aspects remain unclear. Could the authors provide a more detailed explanation of the results shown in Figure 2?
2. Claim about the trade-off between performance and cost. The submission asserts that pathway protection maintains high performance while reducing computational overhead. While the experiments show promising results in terms of performance, a more detailed analysis comparing the computational cost (e.g., memory usage, FLOPs)  would provide stronger evidence.

**Essential References Not Discussed:**

This manuscript lacks comparisons with some relevant approaches, such as [1, 2, 3].

[1] "NISPA: Neuro-Inspired Stability-Plasticity Adaptation for Continual Learning in Sparse Networks." International Conference on Machine Learning.
[2] "Spacenet: Make free space for continual learning." Neurocomputing.
[3] "Sparcl: Sparse continual learning on the edge." Advances in Neural Information Processing Systems.

**Experimental Designs Or Analyses:**

Yes. 2. Although multiple datasets were used, no corresponding comparative experiments were conducted on larger datasets, such as ImageNet-R.

**Methods And Evaluation Criteria:**

Yes

**Other Comments Or Suggestions:**

Please check the above concerns.

**Other Strengths And Weaknesses:**

Strengths:
1. The paper explores the lottery ticket hypothesis under neural network sparsity. By analyzing the complete pathway from input to output, it ensures task-specific pathway preservation.
2. To validate the effectiveness of the proposed framework, experiments are conducted on networks of different sizes across datasets of varying complexity.
3. The paper is well-structured and written, enhancing readability and facilitating a deeper understanding of the content.

Weaknesses:
The authors need to conduct experiments on larger datasets (Imagenet-1K?) and backbones (maybe resnet50? vit?).

**Questions For Authors:**

Please check the above concerns.

**Relation To Broader Scientific Literature:**

The method contributes to the broader scientific literature by addressing the continual learning problem with a novel method.

**Theoretical Claims:**

Yes, no issues found

---

> ### Author Rebuttal · Authors · 2025-03-31
>
> ****Response to C1:**** Thank you for the reviewer's reminder. We sincerely apologize for the misunderstanding caused by our negligence. A precise description will be provided in future revisions. The concept of "Activation Level" refers to the average magnitude of the weights obtained after activation in the last layer of the feature extraction phase. We utilize activation levels to measure whether pathways associated with different tasks can be distinguished. Through the left side of Figure 2, we observed that among these channels, in our method, there consistently exists a channel specific to a task. In contrast, for the LwF method, the activation levels of its channels exhibit a phenomenon of intermixing.
>
> ****Response to C2:**** Thanks for pointing out the issue. We would like to explain it as follows.
> - We conducted a corresponding analysis, including the optimization of our proposed method in terms of time complexity. In the context of our hierarchical matching approach, we analyze its time complexity as follows. Given a deep network with $N_L$ layers, each containing $C$ channels, traditional graph matching incurs a time complexity of O(N^4), where $N$ denotes the total number of nodes in the graph. However, by employing a hierarchical matching strategy for deep networks, we can compute the time complexity separately for each layer and then aggregate the results. Consequently, the overall time complexity of our approach is:
>
> $$ \begin{split} & O(\sum_{1}^{N_L} C^4) = O(\sum_{1}^{N_L} (\frac{N}{N_L})^4) = O(\frac{1}{N_L^3} N^4). \end{split} $$
>
> - Although graph matching is generally an NP-hard problem, using bilateral relaxation transforms it into a solvable problem by converting the general graph matching problem into a bipartite graph matching problem. The original graph matching problem may involve graphs with arbitrary topological structures, but bipartite graph matching is relatively easier to solve computationally. In this paper, we use the Sinkhorn algorithm, which is a polynomial-time algorithm.
>
> ****Response to Essential References:**** Thank you for providing these papers. We indeed omitted the comparisons in our manuscript. We will incorporate references to these papers in our study and conduct comparative analyses accordingly.
>
> **Comparison with [1]**:
>
> 1. This method, like ours, focuses on **protecting connection pathways**.
> 2. **The specific connection points protected are different**. [1] focuses on protecting network connections between adjacent layers.
> 3. **The methods for preserving knowledge from previous tasks differ**. [1] protects important connections for previous tasks by freezing them, whereas we use a soft protection method.
> 4. **The number of trainable parameters for the next task is different**. [1] does not train frozen parameters.
> 5. **The protection of data privacy protection is different**. [1] requires storing previous data, whereas our method does not require storage.
>
> **Comparison with [2]**:
>
> 1. **The different approaches for protecting task-related knowledge**. [2] protects task-specific knowledge by compressing and safeguarding neurons that are crucial for particular tasks. In contrast, our method achieves task knowledge protection through matching-based approach.
> 2. **[2] aims to reduce the interference between different tasks, whereas our method achieves knowledge sharing through interference.**
>
> **Comparison with [3]**:
>
> 1. **The different approaches for protecting task-related knowledge**. [3] employs dynamic masking (including for parameters and gradients) to protect task knowledge, whereas our approach utilizes a matching-based method.
> 2. **Our method facilitates the propagation of knowledge, while [3] prohibits it**. [3] employed in this paper utilizes hard masking, which is detrimental to the sharing of common knowledge across different tasks.
> 3. **The protection of data privacy protection is different**. [3] requires storing previous data.
>
> We replicated two of the above methods and integrated them into our framework for comparative analysis.
>
> ResNet32-based model on the CIFAR-100 dataset.
> |Method|5 splits|10 splits|20 splits
> |-|-|-|-
> |NISPA [1]|69.34±0.81|73.41±0.32|76.42±0.27
> |Sparcl [3]|72.04±0.48|75.08±0.81|77.40±0.21
> |**Ours**|**76.10±0.33**|**81.12±0.90**|**83.19±0.35**
>
> ****Response to Weaknesses:**** We conducted corresponding experimental tests on the ImageNet-R dataset and also performed tests using ResNet50.
>
> **(1) ResNet32-based model on the Imagenet-R.**
> |Method|5 splits|10 splits|20 splits|
> |-|-|-|-|
> |EWC|24.6|27.1|29.3
> |LwF|25.0|28.6|30.3
> |RWalk|26.1|28.4|26.6
> |WSN|28.2|30.4|32.1
> |SPG|23.2|24.1|22.7
> |SPU|26.2|27.6|27.1
> |GPM|26.7|27.1|29.3
> |Ours|**32.8**|**34.3**|**37.3**
>
> **(2) ResNet50-based model on the Cifar100**
> |Method|5 splits|10 splits|20 splits|
> |-|-|-|-|
> |EWC|70.9|62.7|57.6
> |LwF|79.2|80.3|82.7
> |RWalk|70.3|66.3|62.1
> |WSN|80.2|81.4|84.3
> |SPG|55.3|57.2|55.1
> |SPU|60.5|62.1|61.7
> |Ours|**84.4**|**87.5**|**89.7**

---

> > ### Comment · Reviewer_Wbqt · 2025-04-05
> >
> > Thank you for the rebuttal. Most of my concerns have been addressed. I will increase my score accordingly.

---

> > > ### Author Response · Authors · 2025-04-05
> > >
> > > **Many thanks for increasing the score!**
> > >
> > > We sincerely thank the reviewer for the valuable feedback and recognition. We are glad that the additional comparisons and experimental results addressed the concerns and contributed positively to the evaluation. We will incorporate these newly added analyses and experimental results into the final version of the paper to further strengthen the presentation and completeness of our work.

---

### Official Review · Reviewer_EQUe · 2025-03-11

**Overall Recommendation:** 1

**Summary:**

This paper proposes a new framework for continual learning (CL) called Learning without Isolation (LwI), which introduces pathway protection as a mechanism to mitigate catastrophic forgetting. Unlike traditional CL methods that focus on parameter protection, LwI prioritizes preserving activation pathways in deep neural networks, inspired by neuroscientific principles of sparsity in neural activations. The model fusion process is framed as a graph matching problem, where activation pathways in neural networks are aligned using a similarity-based approach. The proposed method is rehearsal-free and the authors evaluate it on CIFAR-100 and Tiny-ImageNet, using ResNet-32 and ResNet-18, demonstrating that the proposed method outperforms the baselines.

**Claims And Evidence:**

Claims with support:

- The authors argue that parameter protection leads to task isolation and inefficient parameter usage. Empirical results show that LwI outperforms parameter isolation methods (e.g., WSN) in both task-aware and task-agnostic settings.

- The paper introduces a graph-matching approach to align activation pathways before merging models. The method achieves better performance, as shown in ablation studies, where removing pathway matching leads to lower accuracy.

Claims that need further support:

- The paper states that graph matching has a complexity of O(N^4), but uses a layer-wise approach to reduce computational cost. No detailed runtime analysis or comparison with other CL methods is provided.

- The paper argues that the proposed method is effective. The proposed method does outperform the baselines, but the accuracies are too weak as there existing methods such as [1] that perform significantly better on both CIFAR-100 and Tiny-ImageNet. For example, [1] is a rehearsal-free method and it achieves more than 65% accuracy on task-agnostic CIFAR-100 10-splits while the proposed method achieves only 30% accuracy.

[1] A theoretical study on solving continual learning. NeurIPS 2022

- The authors claim that parameter-isolation methods need to know the task identity for inference. However, this is not correct. Task incremental learning (TIL) methods such as parameter isolation can be task-agnostic as theoretically demonstrated in [1].

**Essential References Not Discussed:**

Refer to the previous comments

**Experimental Designs Or Analyses:**

The experimental design in the paper is generally well-structured, but there are some areas that could be improved or clarified.

- CIFAR-100 and Tiny-ImageNet are widely accepted benchmarks for continual learning. The datasets are split into 5, 10, and 20 tasks, allowing the evaluation of performance under different levels of granularity.

- Missing comparisons with some recent CIL methods such as [1] and parameter-isolation methods such as SupSup [2].

[1] A theoretical study on solving continual learning. NeurIPS 2022
[2] Supermasks in superposition. NeurIPS 2022

- Task-incremental learning (TIL) methods such as WSN can also be used for task-agnostic evaluation [1].

**Methods And Evaluation Criteria:**

The proposed methods and evaluation criteria largely align with the problem of continual learning.

**Other Comments Or Suggestions:**

There are several minor mistakes.

- ”Activation Level” in lines 087. In latex, use `` for opening a quotation. This incorrect quotation appears in other places throughout the paper.

- "... over-parameterized deep deep networks to allow flexibility for future tasks." The word 'deep' appeared twice

**Other Strengths And Weaknesses:**

NA

**Questions For Authors:**

NA

**Relation To Broader Scientific Literature:**

- Relation to CL: LwI introduces a new paradigm where pathway protection is emphasized over parameter protection, leveraging graph matching for model fusion.

- Relation to model fusion and graph matching: Moves beyond naive weight averaging by using structured pathway alignment, ensuring better knowledge transfer between tasks.

**Theoretical Claims:**

No theoretical analysis was provided. Some arguments made by the authors need theoretical justification. For example, "Simple averaging may lead to interference and even cancellation of effective components, a concern exacerbated during continual learning" in lines 104-107.

---

> ### Author Rebuttal · Authors · 2025-03-31
>
> **Response to C1:** Thank you for the reviewer's reminder.
> - We conducted a corresponding analysis, including the optimization of our proposed method in terms of time complexity. In the context of our hierarchical matching approach, we analyze its time complexity as follows. Given a deep network with $N_L$ layers, each containing $C$ channels, traditional graph matching incurs a time complexity of O(N^4), where $N$ denotes the total number of nodes in the graph (All neurons in the neural network). However, by employing a hierarchical matching strategy for deep networks, we can compute the time complexity separately for each layer and then aggregate the results. Consequently, the overall time complexity of our approach is:
> $$ \begin{split} & O(\sum_{1}^{N_L} C^4) = O(\sum_{1}^{N_L} (\frac{N}{N_L})^4) = O(\frac{1}{N_L^3} N^4). \end{split} $$
> - To address the reviewers' concerns, we tested the time cost and found that due to the addition of the matching fusion module, our method's runtime is second only to LwF.
>
> **Runtime of ResNet32 on CIFAR-100 (min)**
> |Method|5 splits|10 splits|20 splits|
> |-|-|-|-|
> |EWC|157|159|201
> |LwF|105|145|173
> |RWalk|225|236|260
> |WSN|276|298|321
> |Ours|110|157|186
>
> **Response to C2 and Experimental2,3:** Thank you for the reviewer's reminder.
> - During inference, the space and time complexity of [1] are both **O(N)**, where N is the number of tasks. In contrast, our method has both time and space complexity of **O(1)**. To ensure a fair comparison under **O(1)** complexity, we tested using the mask of the last task as well as the intersection of masks from different tasks, ensuring that the time complexity during testing remains **O(1)**. The results of methods such as **WSN** and **GPM** were obtained through comparisons in task-incremental learning.
> **ResNet18-based model on the Cifar100.**
> |Method|10 splits|20 splits|
> |-|-|-|
> |Ours|**50.9**|**47.7**
> |HAT + [1]\(Final)|10.2|12.9
> |HAT + [1]\(Intersection)|13.5|15.4
> |SupSup + [1]\(Final)|15.2|18.8
> |SupSup+ [1]\(Intersection)|18.7|23.4
>
> - Due to differences in the **training environment, dataset splitting methods, and training details** (e.g., we use **200 epochs**, whereas [1] uses **700 and 1000 epochs**), the final training results vary. Additionally, the baseline methods in our paper are implemented based on the code from **[2]**, which aligns well with the corresponding results in **[2]**. We integrated the **LwF** method into the training process of **[1]**, leading to the following results:
> **ResNet18-based model on the Cifar100.**
> |Method|10 splits|20 splits|
> |-|-|-|
> |Ours|**50.9**|**47.7**
> |LwF|42.2|40.8
>
> [2] Class-incremental learning: survey and performance evaluation on image classification. TPAMI,2022.
>
> **Response to C3 and Experimental2,3:** Thank you for the reviewer's reminder.
> Thank you for the reviewer's reminder.
> The reviewer is indeed correct—it is possible to infer the task ID, just not using the approach in [1]. The SupSup paper presents an algorithm for task ID inference based on optimization rather than an O(N) complexity method. However, the inferred task ID may not always be accurate (especially when the number of tasks is large, such as 100 tasks). When we compared this approach with ours, we found that our method outperforms it in task-agnostic scenarios. Lastly, we will revise our wording and include additional experiments in task-agnostic settings.
> |Method|Supsup|Ours
> |-|-|-
> |100 splits|7.47 |**13.63**
>
> **Response to Theory:** We greatly appreciate the reviewer for pointing out this issue. We have performed the corresponding theoretical derivation and will include it in the paper.
>
> The detailed derivation process for 1 and 2 can be found in the response to Reviewer mxBD.
> ### 1. Forgetting Bound
> $$\mathcal{F}(T) \leq \eta\sqrt{2(1-\kappa_S)} \, \text{(Shallow)}+\lambda \epsilon \, \text{(Deep)}+O\left(\sqrt{\frac{\log T}{T}}\right)$$
> ### 2. Task Number
> $$T_{\max} \leq \frac{C}{2\epsilon} \log(1 + \frac{C}{\epsilon \Delta^2})$$
> We divided Tiny-ImageNet into 100 tasks and compared different approaches.
> |Method| SPG |SPU|WSN|EWC|LwF|Supsup|Ours
> |-|-|-|-|-|-|-|-
> |100 splits|43.40|52.90|60.60|50.61|56.82|49.47|**64.63**
>
> ### 3. Naive Avg Bound:
> $$\mathcal{F}_{\text{avg}} \geq \frac{1}{2}\sqrt{\sum \delta_c^2} + \lambda C + \mathcal{O}(1)$$
> - $\mathcal{F}_{\text{avg}}$: Forgetting measure (naive averaging)
> - $\delta_c$: Channel misalignment distance for filter $c$
> When channel alignment quality $\kappa_S > 0.5$ and overlap ratio $\epsilon < 1/C$, the forgetting ratio satisfies:
> $$
> \frac{F_{\text{LwI}}}{F_{\text{avg}}} \approx \sqrt{2(1-\kappa_S)} \, \text{(alignment gain)} \cdot \epsilon \, \text{(sparsity gain)} + O(T^{-1/2})
> $$
> 1. Our method provides an **upper bound** on forgetting
> 2. Naive averaging only provides a **lower bound**
> 3. Experiments
> |Method|5 splits|10 splits|20 splits
> |-|-|-|-
> |Naive Avg|72.13 |74.80 |75.30
> |Ours| **81.10** | **84.90** |**86.49**

---

### Official Review · Reviewer_mxBD · 2025-03-21

**Overall Recommendation:** 3

**Summary:**

The paper introduces a novel approach to continual learning, termed "Learning without Isolation" (LwI), which aims to mitigate catastrophic forgetting by protecting distinct activation pathways for different tasks within a deep network. The key idea is to allocate unique pathways for each task, ensuring that knowledge from previous tasks is preserved while learning new tasks. The authors propose a data-free continual learning method based on graph matching, which aligns channels in the deep network before model fusion. This approach is inspired by the sparsity of activation channels in neural networks and the hierarchical structure of the brain. The method is evaluated on CIFAR-100 and Tiny-Imagenet datasets, demonstrating superior performance compared to existing continual learning methods, particularly in task-agnostic scenarios.

**Claims And Evidence:**

The claims made in the submission are generally supported by clear and convincing evidence, particularly through extensive experimental results on CIFAR-100 and Tiny-Imagenet datasets, which demonstrate the method's superiority over state-of-the-art continual learning approaches. The ablation studies further validate the effectiveness of pathway protection and graph matching. However, some claims, such as the scalability to larger models and the theoretical foundations of pathway protection, lack sufficient evidence. The paper primarily validates the method on smaller models (ResNet32, ResNet18), and a deeper theoretical analysis is absent. Addressing these gaps by including experiments on larger models and providing more theoretical insights would strengthen the overall credibility of the claims.

**Essential References Not Discussed:**

No

**Experimental Designs Or Analyses:**

Please refer to Claims And Evidence part

**Methods And Evaluation Criteria:**

The proposed methods and evaluation criteria are well-suited for addressing the problem of continual learning, particularly in mitigating catastrophic forgetting. The use of benchmark datasets like CIFAR-100 and Tiny-Imagenet, along with task-agnostic and task-aware evaluations, provides a robust framework for assessing the method's performance.

**Other Comments Or Suggestions:**

No

**Other Strengths And Weaknesses:**

## Paper Strengths
The paper presents a novel and theoretically grounded approach to continual learning by leveraging pathway protection and graph matching. This is a significant departure from traditional methods that rely on regularization, rehearsal, or dynamic architectures.

The proposed method does not require storing data from previous tasks, which is a significant advantage in terms of data privacy and storage efficiency.

The authors provide extensive experimental results showing that their method outperforms several state-of-the-art continual learning methods, particularly in task-agnostic settings. The results are convincing and well-supported by ablation studies.

The paper effectively leverages the sparsity of activation channels in deep networks, drawing parallels with neuroscience to justify the approach. This adds a layer of biological plausibility to the method.


## Major Weaknesses
While the empirical results are strong, the paper lacks a thorough theoretical analysis of why the proposed method works. For instance, the authors could provide more insights into the conditions under which pathway protection is most effective or how the method scales with the number of tasks.

The paper primarily validates the method on relatively small models (ResNet32, ResNet18). It would be beneficial to see how the method performs on larger models, such as those used in modern large-scale applications.

**Questions For Authors:**

1. Why is the model limited to the ResNet18 and ResNet32?

**Relation To Broader Scientific Literature:**

The proposed method of pathway protection via graph matching builds on prior work in regularization-based, rehearsal-based, and architecture-based continual learning approaches.

**Theoretical Claims:**

The authors analyze one layer of a deep network channel, and first-order Taylor expansion is used for analysis. It's been correctly demonstrated.

---

> ### Author Rebuttal · Authors · 2025-03-31
>
> **Response to W1:** Thank you for the reviewer's reminder. We have supplemented our theoretical derivations.
> ## 1. Core Theoretical Framework
> ### 1.1 Shallow Layers (Shared knowledge)
> **Core:** Minimize weight differences through optimal transport alignment of similar channels
> 1. Parameter Update
> Merged shallow weights:
> $$W_S^{\text{merged}} = (1-\eta)W_S^{\text{old}} + \eta P_S W_S^{\text{new}}$$
> where $P_S$ is the permutation matrix maximizing similarity.
> 2. Difference Bound
> - Define channel similarity matrix $K_S$ where:
> $$
> P_S = \arg\max_P \text{tr}(P^\top K_S)
> \quad \text{where} \quad
> K_S[i,j] = \frac{w_i^\top w_j}{\|w_i\|\|w_j\|},
> w_i = W_S^{\text{old}}[:,i],
> w_j = W_S^{\text{new}}[:,j]
> $$
> - Optimal transport guarantees existence of $\kappa_S = \max K_S[i,j]$:
> $$\|P_S W_S^{\text{new}} - W_S^{\text{old}}\|_F^2 = 2(1 - \kappa_S)$$
> - Therefore:
> $$\|W_S^{\text{merged}} - W_S^{\text{old}}\| \leq \eta \sqrt{2(1 - \kappa_S)}$$
> ### 1.2 Deep Layers (Task-specific isolation):
> **Core:** Limit inter-task interference through sparse channel separation
> 1. Channel Overlap
> Let tasks $t$ and $t'$ share $\epsilon C$ channels where:
> $$|\mathcal{A}_t \cap \mathcal{A}_{t'}| \leq \epsilon C$$
> 2. Probability Bound
> Using Chernoff bound:
> $$\mathbb{P}(\text{Overlap} \geq \epsilon C) \leq e^{-\epsilon C/2}$$
> 3. **Worst-case Gradient Conflict**
> With Lipschitz constant $\lambda$ and task separation $\Delta$:
> $$\| \nabla L_t - \nabla L_{t'} \| \leq \lambda / (\epsilon \Delta^2)$$
> **Simplified Form**:
> When $\Delta \propto 1/\epsilon$ (implied by sparsity):
> $$
> \text{Deep Term} = \lambda \epsilon
> $$
> ### 1.3 Dynamic Error Term Derivation
> **Core:** Azuma-Hoeffding Inequality for martingales
> 1. Regret Definition
> For task sequence with losses $L_t(\theta_t)$:
> $$R(T) = \sum_{t=1}^T L_t(\theta_t) - \min_\theta \sum_{t=1}^T L_t(\theta)$$
> 2. Concentration Bound
> With probability $1-\delta$:
> $$R(T) \leq \sqrt{2T \log(1/\delta)}$$
> 3. Per-Task Error
> Setting $\delta = 1/T$:
> $$\frac{R(T)}{T} = \mathcal{O}\left(\sqrt{\frac{\log T}{T}}\right)$$
> ### 1.4 Forgetting Bound Theorem
> $$\mathcal{F}(T) \leq \eta\sqrt{2(1-\kappa_S)} \, \text{(Shallow)} + \lambda \epsilon \, \text{(Deep)} + O\left(\sqrt{\frac{\log T}{T}}\right)$$
> Where:
> - $\kappa_S = \max \mathbf{K}_S[i,j]$ (Peak shallow similarity)
> - $\epsilon$ = Channel overlap ratio
> - $\lambda$ = Task conflict intensity (Lipschitz constant)
> ### 1.5 Hyperparameter Settings
> |Parameter|Effect on Forgetting|
> |-|-|
> |$\eta$|$\eta \downarrow \Rightarrow \mathcal{F}_S \downarrow$|
> |$\epsilon$|$\epsilon \downarrow \Rightarrow \mathcal{F}_D \downarrow$|
> |$\kappa_S$|$\kappa_S \uparrow \Rightarrow \mathcal{F}_S \uparrow$
>
> ## 2. Task Capacity Bound
> 1. Define regret $R(T) = \sum_{t=1}^T L_t(\theta_t) - L_t(\theta^*)$
> 2. Apply Azuma-Hoeffding:
> $$ \mathbb{P}(R(T) \geq \sqrt{2T \log T}) \leq 1/T $$
> 3. Per-task average: $\frac{R(T)}{T} = \mathcal{O}(\sqrt{\log T/T})$
> 4. Expected overlap: $\mathbb{E}[X] = \epsilon^2 C$
> Concentration:
> $$ \mathbb{P}(X \geq \epsilon C) \leq e^{-\epsilon^2 C/2}$$
> 5. Conflict bound: $\lambda \epsilon C$
> 6. Channel combinations: $\binom{C}{\epsilon C} \geq T$
> 7. Stirling approximation:
> $$ \log T \leq C H(\epsilon) $$ where $H(\epsilon)$ is binary entropy
> 8. Final bound:
> $$ T_{\max} \leq \frac{C}{2\epsilon} \log(1 + \frac{C}{\epsilon \Delta^2}) $$
> ## 3. Experiments
> We divided Tiny-ImageNet into 100 tasks and compared different approaches.
>
> |Method| SPG |SPU|WSN|EWC|LwF|Supsup|Ours
> |-|-|-|-|-|-|-|-
> |100 splits|43.40|52.90|60.60|50.61|56.82|49.47|**64.63**
>
> **Response to W2:** Thank you for the reviewer's suggestions.
> - At first, we considered that CL is designed for resource-constrained edge devices (e.g., mobile phones) where models must adapt quickly to new tasks with limited computation/memory.
> - To address the reviewer's concern. We conducted corresponding experimental tests on the ImageNet-R dataset and also performed tests using ResNet50.
>
> **(1) ResNet32-based model on the Imagenet-R**
> |Method|5 splits|10 splits|20 splits|
> |-|-|-|-|
> |EWC|24.6|27.1|29.3
> |LwF|25.0|28.6|30.3
> |RWalk|26.1|28.4|26.6
> |WSN|28.2|30.4|32.1
> |SPG|23.2|24.1|22.7
> |SPU|26.2|27.6|27.1
> |GPM|26.7|27.1|29.3
> |Ours|**32.8**|**34.3**|**37.3**
>
> **(2) ResNet50-based model on the Cifar100**
> |Method|5 splits|10 splits|20 splits|
> |-|-|-|-|
> |EWC|70.9|62.7|57.6
> |LwF|79.2|80.3|82.7
> |RWalk|70.3|66.3|62.1
> |WSN|80.2|81.4|84.3
> |SPG|55.3|57.2|55.1
> |SPU|60.5|62.1|61.7
> |Ours|**84.4**|**87.5**|**89.7**
>
> **Response to Q1:** Thanks for the comments. We would like to explain them as follows:
> - When the number of tasks grows or data complexity rises, the model requires more channels to mitigate inter-task interference.
> - ResNet18's deep channels (512) far exceed those of ResNet32 (64), enabling ResNet18 to better meet the demands of sparse allocation in complex task scenarios. We verified that, under the same scenario, the sparser the deeper layers of the network, the greater the performance improvement.

---

> > ### Comment · Reviewer_mxBD · 2025-04-04
> >
> > Thanks for the authors' rebuttal. My concerns have been addressed. I will keep my score and support acceptance.

---

> > > ### Author Response · Authors · 2025-04-05
> > >
> > > We sincerely thank the reviewer for the positive feedback on our work. We would like to provide further supporting evidence here and kindly hope that the reviewer may consider increasing the score.
> > >
> > > **1. Comparison between LwI and naive averaging**
> > >
> > > ##### LwI Bound:
> > > $$
> > > \mathcal{F}_{\text{LwI}} \leq \eta \sqrt{2(1 - \kappa_S)} + \lambda \epsilon C + \mathcal{O}(\sqrt{\frac{2 \log T}{T}})
> > > $$
> > >
> > > ##### Naive Avg Bound:
> > > $$
> > > \mathcal{F}_{\text{avg}} \geq \frac{1}{2}\sqrt{\sum \delta_c^2} + \lambda C + \mathcal{O}(1)
> > > $$
> > >
> > > - $\mathcal{F}_{\text{avg}}$: Forgetting measure (naive averaging)
> > > - $\delta_c$: Channel misalignment distance for filter $c$
> > >
> > > When channel alignment quality $\kappa_S > 0.5$ and overlap ratio $\epsilon < 1/C$, the forgetting ratio satisfies:
> > > $$
> > > \frac{F_{\text{LwI}}}{F_{\text{avg}}} \approx \sqrt{2(1-\kappa_S)} \, \text{(alignment gain)} \cdot \epsilon \, \text{(sparsity gain)} + O(T^{-1/2})
> > > $$
> > >
> > > 1. Our method provides an **upper bound** on forgetting
> > > 2. Naive averaging only provides a **lower bound**
> > > 3. Experiments
> > >
> > > |Method|5 splits|10 splits|20 splits
> > > |-|-|-|-
> > > |Naive Avg|72.13 ± 0.98|74.80 ± 1.65|75.30 ± 1.27
> > > |Ours|**81.10 ± 0.80**|**84.90 ± 0.36**|**86.49 ± 0.55**
> > >
> > > **2. Improvements from the soft matching algorithm.**
> > >
> > > We obtain the matrix $K$, which represents the ground cost of moving the neurons from the $l$-th layer of model1 to the $l$-th layer of model2.
> > >
> > > The Sinkhorn algorithm is then used to solve for the corresponding transport matrix:
> > >
> > > - Entropy regularization is applied to achieve the soft matching process: $S = exp(-K/\epsilon)$，where $\epsilon$ is the entropy regularization parameter.
> > >
> > > - The iterative process:
> > >
> > >     - Step1: $P = P*(\frac{\mu}{\sum{SP}})^\text{T},$ by iterating with row constraints, we obtain the corresponding $P$, where $(·)^\text{T}$ represents the transpose, and $\mu$ denotes the probability distribution of the neurons in model1, $P_{ij} \leftarrow \frac{P_{ij}}{\sum_j P_{ij}}(\text{the sum of each row is 1})$.
> > >
> > >     - Step2: $P = P*(\frac{\nu}{\sum{S^\text{T}P}})^\text{T},$ where the updated matrix $P$ is obtained by iterating with column constraints，$\nu$ represents the probability distribution of the neurons in model2, and the rule $P_{ij} \leftarrow \frac{P_{ij}}{\sum_i P_{ij}}(\text{the sum of each column is 1})$ ensures that the sum of each column in $P$ equals 1.
> > >
> > > By iteratively applying the above steps—step 1 for row constraints and step 2 for column constraints—we ultimately obtain the corresponding soft permutation matrix $P$.
> > >
> > > - We compared fusion using hard matching (Ours with OT), and the results show that soft matching is more advantageous. ResNet18-based model on the CIFAR-100 dataset, measured under task-aware scenarios.
> > >
> > > |Method|5 splits|10 splits|20 splits
> > > |-|-|-|-
> > > |Ours with OT|75.73±0.58|77.73±0.76|80.60±0.71
> > > |**Ours**|**81.10±0.80**|**84.90±0.36**|**86.49±0.55**
> > >
> > > **3. Optimization of algorithmic complexity**
> > >
> > > - We conducted a corresponding analysis, including the optimization of our proposed method in terms of time complexity. In the context of our hierarchical matching approach, we analyze its time complexity as follows. Given a deep network with $N_L$ layers, each containing $C$ channels, traditional graph matching incurs a time complexity of O(N^4), where $N$ denotes the total number of nodes in the graph (All neurons in the neural network). However, by employing a hierarchical matching strategy for deep networks, we can compute the time complexity separately for each layer and then aggregate the results. Consequently, the overall time complexity of our approach is:
> > >
> > > $$ \begin{split} & O(\sum_{1}^{N_L} C^4) = O(\sum_{1}^{N_L} (\frac{N}{N_L})^4) = O(\frac{1}{N_L^3} N^4). \end{split} $$
> > >
> > > - Although graph matching is generally an NP-hard problem, using bilateral relaxation transforms it into a solvable problem by converting the general graph matching problem into a bipartite graph matching problem. The original graph matching problem may involve graphs with arbitrary topological structures, but bipartite graph matching is relatively easier to solve computationally. In this paper, we use the Sinkhorn algorithm, which is a polynomial-time algorithm.

---

### Decision · Program_Chairs · 2025-05-01

**Decision:**

Accept (poster)

**Comment:**

This paper proposes a new approach for continual learning by performing pathways protection, and not just parameter protection during training, with some good performance overall. The rebuttal period raised intense discussions, and some reviewers increased their scores. At the end, three reviewers lean towards acceptance, while one reviewer did not oppose it clearly during the AC – reviewers discussion phase. Therefore, the AC urges the authors to include all rebuttal discussions and new experiments in the final version of the paper.